# Online Convex Optimization Over Erdős-Rényi Random Networks

**Jinlong Lei**
Tongji University
Shanghai, China
leijinlong@tongji.edu.cn

**Peng Yi**[*]
Tongji University
Shanghai, China
yipeng@tongji.edu.cn

**Yiguang Hong**
Tongji University
Shanghai, China
yghong@iss.ac.cn

**Jie Chen**
Tongji University
Shanghai, China
chenjie@bit.edu.cn

**Guodong Shi,**
The University of Sydney
NSW, Australia
guodong.shi@sydney.edu.au

## Abstract

The work studies how node-to-node communications over an Erdős-Rényi random network influence distributed online convex optimization, which is vital in solving large-scale machine learning in antagonistic or changing environments. At per step, each node (computing unit) makes a local decision, experiences a loss evaluated with a convex function, and communicates the decision with other nodes over a network. The node-to-node communications are described by the Erdős-Rényi rule, where independently each link takes place with a probability $p$ over a prescribed connected graph. The objective is to minimize the system-wide loss accumulated over a finite time horizon. We consider standard distributed gradient descents with full gradients, one-point bandits and two-points bandits for convex and strongly convex losses, respectively. We establish how the regret bounds scale with respect to time horizon $T$, network size $N$, decision dimension $d$, and an algebraic network connectivity. The regret bounds scaling with respect to $T$ match those obtained by state-of-the-art algorithms and fundamental limits in the corresponding centralized online optimization problems, e.g., $\mathcal{O}(\sqrt{T})$ and $\mathcal{O}(\ln(T))$ regrets are established for convex and strongly convex losses with full gradient feedback and two-points information, respectively. For classical Erdős-Rényi networks over all-to-all possible node communications, the regret scalings with respect to the probability $p$ are analytically established, based on which the tradeoff between the communication overhead and computation accuracy is clearly demonstrated. Numerical studies have validated the theoretical findings.

## 1 Introduction

The online convex optimization paradigm has become a central and canonical solution for machine learning where data is generated sequentially over time, e.g., online routing, ad. selection for search engines, and spam filtering ([1–4]). Instead of attempting to model the dynamical data, which is often not possible due to fundamental complexity and efficiency challenges, an online convex optimization framework adapts decisions along with the arrival of unforeseen data. After committing to a decision, a convex loss is incurred that is unknown beforehand and may vary over time. There are two basic types of online convex optimization settings in terms of the knowledge that the learner possesses

---

[*]The first two authors contributed equally to the work.

over the loss: with the full gradient feedback, the learner has access to a gradient oracle of the loss function; with the bandit feedback, the learner only observes the losses at points around the decisions.

The goal of the learner is then to minimize its regret by adapting the decisions along the streaming data, measured by the difference between the cumulative loss of online decisions and the loss of the best decision chosen in hindsight. How the regret scales in a finite time horizon $T$ with problem parameters is a central theme in studies of different algorithms. The gradient descent algorithm was proved to guarantee regret bounds $\mathcal{O}(\sqrt{T})$ and $\mathcal{O}(\ln(T))$ for convex and strongly convex loss functions ([5, 6]), respectively, which were later shown to be minimax optimal ([7, 8]).

Distributed online convex optimization ([9, 10]) is preferable in learning over networks when the streaming data are collected at multiple nodes (e.g., sensor networks, smart phones or personal wearable devices). In distributed online convex optimization, the nodes commit to local decisions and then experience local losses that are unknown to the other nodes in the network. It turnes out that, by properly sharing information with neighbors, nodes can collaboratively minimize the accumulated system-wide loss and achieve regret bounds comparable to the centralized case.

## 1.1 The Framework

Consider $N$ nodes indexed in the set $\mathcal{V} = \{1, \ldots, N\}$. At time $t$, node $i \in \mathcal{V}$ makes a decision $\mathbf{x}_{i,t} \in \mathcal{K}$ with $\mathcal{K} \in \mathbb{R}^d$ being a convex set. As a learner, each node can have the following two types of loss information: (i) In the full information feedback, a loss function $f_{i,t}$ is revealed to node $i$ at time $t$; (ii) In the bandit feedback, the function value of $f_{i,t}$ at one or two points around $\mathbf{x}_{i,t}$ is revealed to node $i$ at time $t$.

The network communication structure is described by a connected and undirected graph $\mathcal{G} = (\mathcal{V}, \mathcal{E})$, which serves as a collection of all possible node-to-node communication channels. At time $t$, an Erdős-Rényi graph $\mathrm{G}_t = (\mathcal{V}, \mathrm{E}_t)$ is generated over the prescribed graph $\mathcal{G}$, where independently with time and other links, $\{i, j\} \in \mathrm{E}_t$ with a probability $0 < p < 1$ for all $\{i, j\} \in \mathcal{E}$. Note that in classical Erdős-Rényi graphs [11], $\mathcal{G}$ is a complete graph. Here we allow $\mathcal{G}$ to be any connected graph, for the sake of presenting a general online learning framework over networks, when all-to-all communications might not exist in practice. Some realizations of classical Erdős-Rényi graphs with $N = 300$ and $p = 0.006$ are shown as follows:

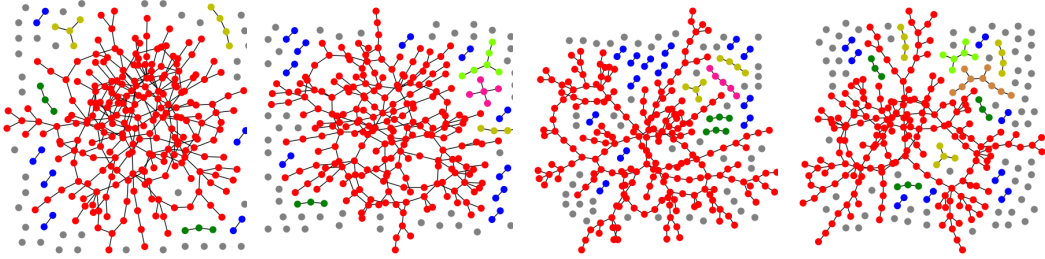

| (a) $\mathrm{G}_t$ at $t = 1$ | (b) $\mathrm{G}_t$ at $t = 2$ | (c) $\mathrm{G}_t$ at $t = 3$ | (d) $\mathrm{G}_t$ at $t = 4$ |

The objective is to design distributed online decision learning algorithms so that each node $i$ identifies the decision $\mathbf{x}_{i,t}, t = 1, \ldots, T$ to minimize the accumulated system-wide loss $\sum_{t=1}^{T} \sum_{i=1}^{N} f_{i,t}(\mathbf{x})$. When choosing $\mathbf{x}_{i,t+1}$, node $i$ can utilize its previous local decisions $\mathbf{x}_{i,k}, k \leq t$ and revealed losses $f_{i,k}, k \leq t$, along with the information received from its neighbors over $\mathrm{G}_t$. The performance of a distributed learning algorithm is captured by the regrets at various nodes compared with the best fixed decision in the hindsight $\mathbf{x}^* = \operatorname{argmin}_{\mathbf{x} \in \mathcal{K}} \sum_{t=1}^{T} \sum_{i=1}^{N} f_{i,t}(\mathbf{x})$. The regret of node $j \in \mathcal{V}$ is thus

$$\mathsf{Reg}(j, T) = \sum_{t=1}^{T} \sum_{i=1}^{N} f_{i,t}(\mathbf{x}_{j,t}) - \sum_{t=1}^{T} \sum_{i=1}^{N} f_{i,t}(\mathbf{x}^*). \tag{1}$$

The motivation to study online convex optimization over Erdős-Rényi graphs is as follows. Firstly, the distributed online learning can happen in Internet of things or social networks, while the Erdős-Rényi

graph is one of the fundamental models for network communications or social interactions (see [12, 13]), where information channels are represented by independent links that are either on with a probability $p$ or off with a probability $1-p$. Therefore, the performance of online convex optimization over Erdős-Rényi graphs would serve as a benchmark for online learning over practical networks. Secondly, in the dense network, we tend to avoid communicating along each edge per iteration to decrease communication, for which Erdős-Rényi graphs can be viewed as a strategic way to organize node-to-node communications. As such, the probability $p$ becomes a design parameter for tuning the trade-off between the communication overhead and the computation accuracy. It has been found that Erdős-Rényi random graphs can outperform fully connected graphs in some distributed training tasks ([14]). Finally, Erdős-Rényi graphs allow us to intuitively have a deep theoretical characterization of how the graph topology and connectivity probability influence the network regret of online convex optimization, and paves the way to study distributed online convex optimization over other complex graph models.

## 1.2 Main Results

We consider online distributed gradient descents under Erdős-Rényi graphs, and establish the regret bounds explicitly in term of time horizon $T$, the underlying graph $\mathcal{G}$, the probability $p$, and the decision complexity $d$.

**Algorithms:** The node adapts its decision with gradient descents and local averaging over Erdős-Rényi graphs. In the full information case, the local gradient $\nabla f_{i,t}(\mathbf{x}_{i,t})$ is utilized first, and then the decision is made by averaging across its neighboring information and projecting onto a feasible set. When the nodes can only observe loss function values, the one-point bandit or two-points bandit around the current decision is used to get a randomized approximation of the gradient.

**Regret Bounds:** In the full gradient case, the algorithm achieves the regrets $\mathcal{O}(\sqrt{T})$ and $\mathcal{O}(\ln(T))$ for convex and strongly convex losses, respectively. The bounds $\mathcal{O}(d^{1/2}T^{3/4})$ (convex) and $\mathcal{O}(d^{2/3}T^{2/3}\ln^{1/3}(T))$ (strongly convex) are achieved in the one-point bandit case, while the bounds are improved to $\mathcal{O}(d\sqrt{T})$ and $\mathcal{O}(d^2\ln(T))$ in the two-points bandit case. It is shown that the regrets scale with network size $N$ by a magnitude of $N^c$ where $c$ takes $\frac{3}{2}$, $\frac{5}{4}$, or $\frac{7}{6}$, depending on the convexity or strong convexity of the losses, and the information feedback type. In term of time horizon $T$, the regret scalings match those obtained by state-of-the-art algorithms in the centralized online convex optimization, and distributed online convex optimization with deterministic communications. In addition, for classical large-scale Erdős-Rényi networks over all-to-all possible node communications, the regret bounds scale with the probability $p$ by a magnitude of $p^c$ where $c = -1, -1/2$, or $-1/3$. The regret bounds along with the communication complexity over classical Erdős-Rényi graphs are shown in Table 1, while the communication complexity is measured by the expected rounds of node-to-node communications taken by each node over a time horizon $T$.

| Settings | Regrets for convex losses | Regrets for strongly convex losses | Communication Complexity |
|---|---|---|---|
| Full information | $O(p^{-1}N^{3/2}\sqrt{T})$ | $\mathcal{O}(p^{-1}N^{3/2}\ln(T))$ | $pNT$ |
| One-point bandit | $\mathcal{O}(p^{-1/2}d^{1/2}N^{5/4}T^{3/4})$ | $\mathcal{O}(p^{-1/3}d^{2/3}N^{7/6}T^{2/3}\ln^{1/3}(T))$ | $pNT$ |
| Two-points bandit | $\mathcal{O}(p^{-1}dN^{3/2}\sqrt{T})$ | $\mathcal{O}(p^{-1}d^2N^{3/2}\ln(T))$ | $pNT$ |

Table 1: Regret bounds and communication complexity over classical Erdős-Rényi graphs

## 1.3 Related Work

The early works [5] and [6] about the centralized online convex optimization in the full information feedback obtained regrets $\mathcal{O}(\sqrt{T})$ and $\mathcal{O}(\ln(T))$ for convex and strongly convex losses, respectively. With one-point bandit feedback, the seminal work [15] modified the gradient descent algorithm by replacing the gradient with a randomized estimate, and showed that the expected regret was $\mathcal{O}(T^{3/4})$ for bounded and Lipschitz-continuous convex losses, whereas the regret $\mathcal{O}(T^{2/3})$ was obtained in [16] for the setting of Lipschitz and strongly convex losses. It remains an open problem to design an optimal algorithm for the one-point bandit online convex optimization, whereas [17] proved that the optimal regret can not be better than $\Omega(\sqrt{T})$ even for strongly convex losses. In some special cases, the minimax regret is exactly $\mathcal{O}(\text{poly}(\ln(T))\sqrt{T})$ , e.g., the losses are Lipschitz and linear [17, 18],

or they are both smooth and strongly-convex [19]. The recent work [20] designed some kernel-based methods with $\mathcal{O}(\text{poly}(\ln(T))\sqrt{T})$ regret and polynomial computing time. [16] extended the one-point bandit feedback to the multi-points bandit feedback where loss can be observed at multiple points around the decision, and established the expected regret bounds $\mathcal{O}(\sqrt{T})$ and $\mathcal{O}(\ln(T))$ for convex and strongly convex losses. In this work, we design distributed algorithms over random graphs with full gradient feedback, one-point bandit feedback, and two-points bandit feedback, which can recover the regret bounds in the centralized methods ([6], [15] and [16]).

Distributed online convex optimization has drawn much research attention in online learning over networks. [21] proposed distributed regression algorithms for distributed online linear regression, and established the regret bound $\mathcal{O}(T^{3/4})$ when the decision set is unbounded and $\mathcal{O}(\sqrt{T})$ in case of bounded decision set. [22] proposed a distributed subgradient algorithm that achieved the regrets $\mathcal{O}(\sqrt{T})$ and $\mathcal{O}(\ln(T))$ for convex and strongly convex losses, while [23] introduced a distributed online dual subgradient averaging with a regret $\mathcal{O}(\sqrt{T})$ for convex losses. The aforementioned works [21–23] considered the static and strongly-connected balanced directed networks. [24] designed a distributed online primal-dual algorithm with the regrets $\mathcal{O}(\ln(T))$ (or $\mathcal{O}(\sqrt{T})$) for strongly convex (or convex) losses when the communication networks are jointly connected. While [25] considered distributed online convex optimization with long-term constraints over jointly connected networks, and proposed a decentralized algorithm with regret and cumulative constraint violation matching the fundamental limits in the corresponding centralized online optimization problem. In addition, [26] proposed a distributed online subgradient push-sum algorithm for distributed online convex optimization over unbalanced time-varying digraphs, and obtained a regret $\mathcal{O}\big((\ln^2(T))\big)$ for strongly convex losses. [27] further deigned a consensus-based primal-dual method for distributed online convex optimization with global coupling constraints, and obtained a regret $\mathcal{O}(\sqrt{T})$ for weight-balanced and jointly connected networks. To replace the expensive projection operation with a simpler linear optimization, [9] proposed a distributed online conditional gradient method that achieved a regret $\mathcal{O}(T^{\frac{3}{4}})$ for convex losses. To mitigate the impact of slow nodes in synchronized distributed stochastic online convex optimization, [10] fixed the computation time of minibatch gradients for each node per step, and proved a regret $\mathcal{O}(\sqrt{\bar{m}})$, where $\bar{m}$ is the expected total number of gradient samples used up to time $T$. It is worth mentioning that all aforementioned works are restricted to fixed or deterministically switching graphs. This work explores the distributed online convex optimization over Erdős-Rényi graphs and obtains the same regret bounds as previous distributed online convex optimization schemes, e.g., [22] and [24], while further characterizes how the regrets are influenced by the link probability $p$ of the Erdős-Rényi graphs (or the algebraic network connectivity).

## 2 Full Information Feedback

This section focuses on the full information feedback, where each node $i$ has access to the gradient of the loss function $f_{i,t}$ at the query point. We consider the following assumptions on the constraint set and the loss functions, which are also used in the existing literature, see e.g., [1, 28].

**Assumption 1.** *The convex set $\mathcal{K}$ is compact, i.e., there exists $D_1$ such that*

$$\|\mathbf{x} - \mathbf{y}\| \leq D_1, \quad \forall \mathbf{x}, \mathbf{y} \in \mathcal{K}.$$

**Assumption 2.** *For each $i \in \mathcal{V}$ and all $t = 1, \cdots, T$, the loss function $f_{i,t}$ is convex with bounded gradients over $\mathcal{K}$:*

$$\max_{i \in \mathcal{V}} \max_{t=1,\cdots,T} \max_{\mathbf{x} \in \mathcal{K}} \|\nabla f_{i,t}(\mathbf{x})\| \leq G_f.$$

**Assumption 3.** *For each $i \in \mathcal{V}$ and $t = 1, \ldots, T$, $f_{i,t}$ is $\alpha$-strongly convex over $\mathcal{K}$, i.e.,*

$$f_{i,t}(\mathbf{x}) - f_{i,t}(\mathbf{y}) \geq \big(\mathbf{x} - \mathbf{y}\big)^T \nabla f_{i,t}(\mathbf{y}) + \frac{\alpha}{2}\|\mathbf{x} - \mathbf{y}\|^2, \quad \forall \mathbf{x}, \mathbf{y} \in \mathcal{K}.$$

Let the neighbor set of node $i$ with the prescribed graph $\mathcal{G}$ be denoted as $\mathcal{N}_i \triangleq \{j \in \mathcal{V} : \{i, j\} \in \mathcal{E}\}$. Denote by $\mathbf{L}$ the Laplacian matrix of the graph $\mathcal{G}$, where $[\mathbf{L}]_{ij} = -1$ if $\{i, j\} \in \mathcal{E}$, $[\mathbf{L}]_{ii} = |\mathcal{N}_i|$, and and $[\mathbf{L}]_{ij} = 0$, otherwise. Here and thereafter, $|\cdot|$ stands for the cardinality of a set. At each time $t$, node $i$ can receive the decisions of its neighbors in the random set $\mathrm{N}_{i,t} \triangleq \{j \in \mathcal{V} : \{i, j\} \in \mathrm{E}_t\} \subseteq \mathcal{N}_i$. Denote by $\mathbf{L}_t$ the Laplacian matrix of the graph $\mathrm{G}_t$, and $\mathbf{I}_N$ the $N \times N$ identity matrix.

Denote $\rho \triangleq \sqrt{\text{esp}\left(\mathbf{I}_N - 2ap\mathbf{L} + a^2\mathbb{E}[\mathbf{L}_t^2]\right)} < 1$ with $\text{esp}(\mathbf{P})$ denoting the essential spectral radius of a stochastic matrix $\mathbf{P}$: $\text{esp}(\mathbf{P}) = \max\{|\lambda| : \lambda \text{ is the eigenvalue of } \mathbf{P} \text{ different from } 1\}$. The parameter $\rho$ reflects the algebraic connectivity of Erdős-Rényi graphs. Qualitatively, $\rho$ is small when the underlying graph $\mathcal{G}$ has a good connectivity or the probability $p$ is large.

We generalize the algorithm of [5] to a distributed setting, and propose Algorithm 1 for online decision making with local gradients and neighboring information over Erdős-Rényi graphs.

---

**Algorithm 1** Distributed online gradient descent with full gradients

---

Input: convex set $\mathcal{K}$, $T$, and step-sizes $\{\eta_t\}$.
Initialize: Set $\mathbf{x}_{i,1} = \mathbf{0}$ for each node $i \in \mathcal{V}$.
1: for $t = 1$ to $T$ do
2:     Node $i$ commits to a decision $\mathbf{x}_{i,t}$, receives the loss $f_{i,t}$, and computes

$$\mathbf{y}_{i,t} = \mathbf{x}_{i,t} - \eta_t \nabla f_{i,t}(\mathbf{x}_{i,t}). \tag{2}$$

3:     Nodes $i$ communicates $\mathbf{y}_{i,t}$ to its neighbors over $\mathrm{G}_t$ and updates its decision as follows

$$\mathbf{x}_{i,t+1} = \Pi_{\mathcal{K}}\left(\left(1 - a|\mathrm{N}_{i,t}|\right)\mathbf{y}_{i,t} + a\sum_{j\in\mathrm{N}_{i,t}} \mathbf{y}_{j,t}\right), \tag{3}$$

where $0 < a \leq \frac{1}{1+\max_i |\mathcal{N}_i|}$, and $\Pi_{\mathcal{K}}(\mathbf{x})$ denotes the Euclidean projection of a vector $\mathbf{x}$ onto a set $\mathcal{K}$, i.e., $\Pi_{\mathcal{K}}(\mathbf{x}) = \text{argmin}_{\mathbf{y}\in\mathcal{K}} \|\mathbf{x} - \mathbf{y}\|$.
4: end for

---

Next, we establish the expectation-valued regret bounds of Algorithm 1 for convex and strongly convex losses, respectively.

**Theorem 1** (Expected regret for convex losses)**.** *Let Assumptions 1 and 2 hold. Consider Algorithm 1 with $\eta_t = \frac{D_1}{G_f\sqrt{t}}$. Then for each $j \in \mathcal{V}$,*

$$\mathbb{E}\big[\text{Reg}(j,T)\big] \leq 3ND_1G_f\left(0.5 + \frac{2\rho(1+\sqrt{N})}{1-\rho}\right)\sqrt{T}. \tag{4}$$

**Theorem 2** (Expected regret for strongly convex losses)**.** *Let Assumptions 1, 2, and 3 hold. Consider Algorithm 1 with $\eta_t = \frac{1}{\alpha t}$. Then for each $j \in \mathcal{V}$,*

$$\mathbb{E}\big[\text{Reg}(j,T)\big] \leq \frac{NG_f^2}{2\alpha}\left(1 + \frac{6\rho(1+\sqrt{N})}{1-\rho}\right)(1 + \ln(T)). \tag{5}$$

The results are proved according to the following. We firstly establish an upper bound on the node regrets, which depend on the step-sizes and the consensus error across the node decisions. Then by utilizing the properties of Erdős-Rényi graphs, we bound the expected consensus error with the step-sizes and the inverse spectral gap of the expected network. Finally, with appropriately chosen step-sizes, we obtain the regret bounds established in Theorems 1 and 2.

Theorems 1 and 2 show that, for distributed online convex optimization over Erdős-Rényi graphs, the regrets $\mathcal{O}(\sqrt{T})$ and $\mathcal{O}(\ln(T))$ are obtained for convex and strongly convex losses, which are the same as the centralized regrets established in [5] and [6]. For a single agent case, the regrets become $\frac{3D_1G_f}{2}\sqrt{T}$ [1, Theorem 3.1] and $\frac{G_f^2}{2\alpha}(1 + \ln(T))$ [1, Theorem 3.3] for convex and strongly convex losses, respectively. For the multiple nodes case with $N \geq 2$, the results explicitly show how the regrets depend on the network size and the algebraic network connectivity. Both Theorems 1 and 2 imply that the average regret (divided by $N$) increases with $N$. This is possibly because the increasing node number would increase the nodes' information heterogeneity and make the network regret minimization more difficult. In addition, the derived regrets showed the inverse dependence on the spectral gap of the expected network, which is quite natural since it is well-known to determine the mixing rates in random walks on graphs, and the information propagation over Erdős-Rényi graphs is closely tied to the random walk on the expected network.

Let the underlying $\mathcal{G} = \{\mathcal{V}, \mathcal{E}\}$ be a complete graph to recover the classical Erdős-Rényi graphs. It was shown in [29, Example 4.7] that $\rho^2 = (1-p)\left(1 - p + 2p\frac{N-1}{N^2}\right)$. Then for a sufficiently large $N$, $\frac{\rho}{1-\rho}$ is approximately $p^{-1} - 1$. Note that the expected number of node-to-node communications of each node over a time horizon is $pNT$, while the expected regrets for convex and strongly convex losses are approximately $\mathcal{O}\left(p^{-1}N^{3/2}\sqrt{T}\right)$ and $\mathcal{O}\left(p^{-1}N^{3/2}\ln(T)\right)$, which decrease as $p$ increases. It is seen that the increasing of $p$ reduces the regret bounds while increases the communication overhead. Therefore, for a fixed time horizon $T$ and a fixed prescribed graph $\mathcal{G}$, the parameter $p$ can be used to balance the communication overhead and the computation accuracy.

## 3 One-point Bandit Feedback

This part considers the one-point bandit feedback, when each node $i$ can only observe the value of the loss function $f_{i,t}$ at a single point around $\mathbf{x}_{i,t}$. Motivated by [15], we replace $\nabla f_{i,t}(\mathbf{x}_{i,t})$ in (2) with its randomized estimate $\mathbf{g}_{i,t}$ given by (6) to obtain Algorithm 2. It is shown in [15, Lemma 1] that $\mathbb{E}[\mathbf{g}_{i,t}] = \nabla \hat{f}_{i,t}(\mathbf{x}_{i,t})$, where $\hat{f}_{i,t}(\mathbf{x}) = \mathbb{E}_{\mathbf{u}\in\mathcal{B}}[f_{i,t}(\mathbf{x} + \delta\mathbf{u})]$ with $\mathcal{B} = \{\mathbf{u} \in \mathbb{R}^d : \|\mathbf{u}\| \leq 1\}$. Node $i$ adapts its decision $\mathbf{x}_{i,t+1}$ by (7), where the projection onto $(1-\xi)\mathcal{K}$ is used to guarantee that $\mathbf{x}_{i,t} + \delta\mathbf{u}_{i,t} \in \mathcal{K}$.

---

**Algorithm 2** Distributed online algorithm with one-point bandit feedback

---

Input: Step sizes $\{\eta_t\}$, the exploration and shrinkage parameters $\delta > 0$ and $\xi \in (0,1)$.
Initialize: Set $\mathbf{x}_{i,1} = \mathbf{0}$ for each node $i \in \mathcal{V}$.
1: for $t = 1$ to $T$ do
2:    Node $i$ commits to a decision $\mathbf{x}_{i,t}$ and observes $f_{i,t}(\mathbf{x}_{i,t} + \delta\mathbf{u}_{i,t})$, where $\mathbf{u}_{i,t} \in \mathbb{R}^d$ is uniformly chosen from vectors with a unit norm ($\|\mathbf{u}_{i,t}\| = 1$).
3:    Node $i$ constructs a gradient estimator

$$\mathbf{g}_{i,t} = \frac{d}{\delta}f_{i,t}(\mathbf{x}_{i,t} + \delta\mathbf{u}_{i,t})\mathbf{u}_{i,t}, \tag{6}$$

and computes $\mathbf{y}_{i,t} = \mathbf{x}_{i,t} - \eta_t\mathbf{g}_{i,t}$.
4:    Nodes $i$ communicates $\mathbf{y}_{i,t}$ to its neighbors over $\mathrm{G}_t$, and updates its decision as follows

$$\mathbf{x}_{i,t+1} = \Pi_{(1-\xi)\mathcal{K}}\left(\left(1 - a|\mathrm{N}_{i,t}|\right)\mathbf{y}_{i,t} + a\sum_{j\in\mathrm{N}_{i,t}}\mathbf{y}_{j,t}\right). \tag{7}$$

5: end for

---

We impose the following conditions on $\mathcal{K}$ and the losses by slightly modifying Assumptions 1 and 2. Both are conventional in online bandit optimization ([15, 16]).

**Assumption 4.** *The set $\mathcal{K}$ contains a ball of radius $r$ centered at the origin, and is also contained in a ball of radius $R$, i.e., $r\mathcal{B} \subseteq \mathcal{K} \subseteq R\mathcal{B}$.*

**Assumption 5.** *Each loss function $f_{i,t}$ is convex and Lipschitz continuous in $\mathcal{K}$, i.e., there exists a constant $L_f > 0$ such that for each $i \in \mathcal{V}$ and any $t = 1, \cdots, T$:*

$$\|f_{i,t}(\mathbf{x}) - f_{i,t}(\mathbf{y})\| \leq L_f\|\mathbf{x} - \mathbf{y}\|, \quad \forall\mathbf{x}, \mathbf{y} \in \mathcal{K}.$$

Based on Assumptions 4 and 5, there exists a constant $C > 0$ such that,

$$\max_{i\in\mathcal{V}} \max_{t=1,\cdots,T} \max_{\mathbf{x}\in\mathcal{K}} \|f_{i,t}(\mathbf{x})\| \leq C. \tag{8}$$

**Theorem 3** (Convex losses with the one-point bandit feedback). *Let Assumptions 4 and 5 hold. Consider Algorithm 2, where $\eta_t = \frac{2\delta R}{dC\sqrt{t}}$, $\delta = (c_1/c_2)^{0.5}T^{-0.25}$, and $\xi = \frac{\delta}{r}$ with $c_1 \triangleq 3dRC\left(1 + \frac{4\rho(1+\sqrt{N})}{1-\rho}\right)$ and $c_2 \triangleq 2(L_f + C/r)$. Then*

$$\mathbb{E}\big[\mathrm{Reg}(j,T)\big] \leq 2T^{3/4}N\sqrt{6dRC\left(1 + \frac{4\rho(1+\sqrt{N})}{1-\rho}\right)(L_f + C/r)}, \quad \forall j \in \mathcal{V}.$$

**Theorem 4** (Strongly convex losses with the one-point bandit feedback). *Let Assumptions 3, 4, and 5 hold. Consider Algorithm 2, where $\eta_t = \frac{1}{\alpha t}$, $\delta = \left(\frac{2c_3(1+\ln(T))}{c_2 T}\right)^{1/3}$, and $\xi = \frac{\delta}{r}$ with $c_3 \triangleq \frac{d^2 C^2}{2\alpha}\left(1 + \frac{6\rho(1+\sqrt{N})}{1-\rho}\right)$ and $c_2 \triangleq 2(L_f + C/r)$. Then, for each $j \in \mathcal{V}$,*

$$\mathbb{E}\big[\mathrm{Reg}(j,T)\big] \leq 3N d^{2/3} T^{2/3} (1+\ln(T))^{1/3} \left(\frac{C^2(L_f + C/r)^2}{2\alpha}\left(1 + \frac{6\rho(1+\sqrt{N})}{1-\rho}\right)\right)^{1/3}.$$

Algorithm 2 can achieve the regret $\mathcal{O}\big((\frac{\rho}{1-\rho})^{1/2} d^{1/2} N^{5/4} T^{3/4}\big)$ for convex losses, which matches the bound $\mathcal{O}(T^{3/4})$ obtained by [15] in centralized settings, while, for strongly convex losses, the regret bound can be improved to $\mathcal{O}\big((\frac{\rho}{1-\rho})^{1/3} d^{2/3} N^{7/6} T^{2/3} \ln^{1/3}(T)\big)$. The results explicitly show the influence of time horizon $T$, decision variable dimension $d$, network size $N$, and network connectivity $\rho$ on the regret bounds. It is further noticed that for classical Erdős-Rényi graphs, the regrets respectively scale with $p^{-1/2}$ and $p^{-1/3}$ for convex and strongly convex losses.

## 4 Two-points bandit feedback

This part studies the two-points bandit feedback, where the gradient $\nabla f_{i,t}(\mathbf{x}_{i,t})$ is estimated by evaluating the loss function $f_{i,t}$ at two random points around $\mathbf{x}_{i,t}$. Consider Algorithm 3, where the gradient is estimated by (9). Since the distribution of $\mathbf{u}_{i,t}$ is symmetric, by [15, Lemma 1] we have that $\mathbb{E}[\tilde{\mathbf{g}}_{i,t}] = \nabla \hat{f}_{i,t}(\mathbf{x}_{i,t})$ with $\hat{f}_{i,t}(\mathbf{x}) = \mathbb{E}_{\mathbf{u}\in\mathcal{B}}[f_{i,t}(\mathbf{x} + \delta\mathbf{u})]$.

---

**Algorithm 3** Distributed algorithm with two-points bandit feedback

---

Input: Step sizes $\{\eta_t\}$, the exploration and shrinkage parameters $\delta$ and $\xi \in (0,1)$.
Initialize: Set $\mathbf{x}_{i,1} = \mathbf{0}$ for each node $i \in \mathcal{V}$.
1: for $t = 1$ to $T$ do
2:    Node $i$ commits to a decision $\mathbf{x}_{i,t}$, picks a unit-norm vector $\mathbf{u}_{i,t} \in \mathbb{R}^d$ uniformly at random, and observes the values of the loss function $f_{i,t}$ at two points $\mathbf{y}_{j,t}^1 = \mathbf{x}_{i,t} + \delta\mathbf{u}_{i,t}$ and $\mathbf{y}_{j,t}^2 = \mathbf{x}_{i,t} - \delta\mathbf{u}_{i,t}$.
3:    Node $i$ constructs a gradient estimator

$$\tilde{\mathbf{g}}_{i,t} = \frac{d}{2\delta}(f_{i,t}(\mathbf{x}_{i,t} + \delta\mathbf{u}_{i,t}) - f_{i,t}(\mathbf{x}_{i,t} - \delta\mathbf{u}_{i,t}))\mathbf{u}_{i,t}, \tag{9}$$

and computes $\mathbf{y}_{i,t} = \mathbf{x}_{i,t} - \eta_t \tilde{\mathbf{g}}_{i,t}$.
4:    Nodes $i$ communicates $\mathbf{y}_{i,t}$ to its neighbors over $\mathrm{G}_t$, and updates its decision as follows

$$\mathbf{x}_{i,t+1} = \Pi_{(1-\xi)\mathcal{K}}\left(\big(1 - a|\mathrm{N}_{i,t}|\big)\mathbf{y}_{i,t} + a \sum_{j\in\mathrm{N}_{i,t}} \mathbf{y}_{j,t}\right). \tag{10}$$

5: end for

---

With Algorithm 3, the regret of node $j$ is denoted as

$$\mathcal{R}_2(j,T) \triangleq \sum_{t=1}^{T}\sum_{i=1}^{N} \frac{f_{i,t}(\mathbf{y}_{j,t}^1) + f_{i,t}(\mathbf{y}_{j,t}^2)}{2} - \sum_{t=1}^{T}\sum_{i=1}^{N} f_{i,t}(\mathbf{x}^*).$$

**Theorem 5** (Convex losses with the two-points bandit feedback). *Let Assumptions 4 and 5 hold. Consider Algorithm 3, where $\eta_t = \frac{2R}{dL_f\sqrt{t}}$, $\delta = \frac{1}{\sqrt{T}}$, and $\xi = \frac{\delta}{r}$. Then, for each $j \in \mathcal{V}$,*

$$\mathbb{E}\big[\mathcal{R}_2(j,T)\big] \leq N L_f \sqrt{T}\left(3 + R/r + 3dR + \frac{12d\rho R(1+\sqrt{N})}{1-\rho}\right).$$

**Theorem 6** (Strongly convex losses with the two-points bandit feedback). *Let Assumptions 3, 4, and 5 hold. Consider Algorithm 3 with $\eta_t = \frac{1}{\alpha t}$, $\delta = \frac{\ln(T)}{T}$, and $\xi = \frac{\delta}{r}$. Then, for each $j \in \mathcal{V}$,*

$$\mathbb{E}\big[\mathcal{R}_2(j,T)\big] \leq \frac{N d^2 L_f^2}{2\alpha}\left(1 + \frac{6\rho(1+\sqrt{N})}{1-\rho}\right)(1+\ln(T)) + N L_f\left(3 + \frac{R}{r}\right)\ln(T).$$

Theorems 5 and 6 show that Algorithm 3 can recover the regret bounds $\mathcal{O}(\sqrt{T})$ (convex) and $\mathcal{O}(\ln(T))$ (strongly convex) in the full information case, if each node is allowed to query the losses at two points around the decision. Nevertheless, the bounds in the two-points bandit case depend on the decision dimension $d$, and the constants are larger than those of Theorems 1 and 2. In addition, Algorithm 3 imposes Assumption 4 on the convex set, stronger than Assumption 1 required by Algorithm 1. Specially, for classical Erdős-Rényi graphs, the expected regrets become $\mathcal{O}(p^{-1}dN^{3/2}\sqrt{T})$ and $\mathcal{O}(p^{-1}d^2N^{3/2}\ln(T))$ for convex and strongly convex losses, respectively.

## 5 Numerical Experiments

Erdős-Rényi graphs give us information about complex systems in the real world. A prominent motivating example is distributed online learning through random social interactions for exploiting the streaming but private health data generated from wearable personal tracking device ([30]). Thus, in this section, we examine the empirical performance of the proposed distributed algorithms on the **bodyfat** dataset with 14 features and 252 instances from LIBSVM library [2]. Throughout this section, the empirical results are based on averaging across 20 sample trajectories.

Consider a distributed online regularized linear regression problem formulated as follows:

$$\min_{\mathbf{x}\in\mathbb{R}^d} \sum_{t=1}^{T}\sum_{i=1}^{N}(\mathbf{a}_{i,t}^T\mathbf{x}-b_{i,t})^2+\mu\|\mathbf{x}\|^2, \quad s.t. \quad \mathbf{x}\in\mathcal{K}\triangleq\{\mathbf{x}\in\mathbb{R}^d:\|\mathbf{x}\|\leq R\}, \quad (11)$$

where $\mu\geq 0$ denotes the regularization parameter, the data $(\mathbf{a}_{i,t},b_{i,t})\in\mathbb{R}^d\times\mathbb{R}$ with $d=14$ from **bodyfat** is revealed only to node $i$ at time $t$, and the decision is restricted to a sphere $\mathcal{K}$ with radius $R$.

We first demonstrate the empirical performance of the rescaled maximum regret $\mathsf{SReg}(T)\triangleq\frac{\max_j\ \mathsf{Reg}(j,T)}{T}$ as a function of the time horizon $T$. We set $N=30$, $p=0.2$, and $\mu=0$ in (11) to get convex losses. We run Algorithms 1, 2, and 3, and plot the rescaled maximum regret $\mathsf{SReg}(T)$ versus the time horizon $T$ in Figure 1. In addition, we set $\mu=1$ in (11) to construct strongly convex losses, run Algorithms 1, 2, and 3, and plot $\mathsf{SReg}(T)$ versus the time horizon $T$ in Figure 2. It is seen from Figure 2 that the performance gets deteriorated in the first few steps, maybe because there is not enough accumulated data to adapt a good solution in the one-point bandit case. The empirical results shown in both Figure 1 and Figure 2 are accordant with the theoretical results that the time averaged regret goes to zero as $T$ goes to infinity. From both Figures 1 and 2 we can see that the performance of the one-bandit feedback is the worst among the three cases, while the two-points bandit feedback can significantly improve the algorithm performance and is almost comparable with the full information case.

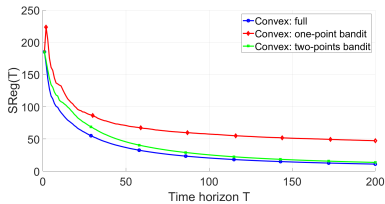

Figure 1: SReg vs time for convex losses.

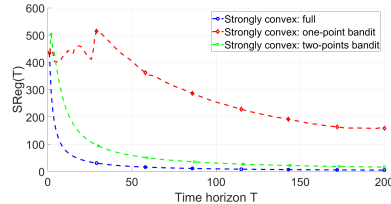

Figure 2: SReg vs time for strongly convex losses.

Next, we fix the time horizon $T=200$ and demonstrate the empirical performance of the maximum regret $\mathsf{MReg}(T)\triangleq\max_j\ \mathsf{Reg}(j,T)$ versus the link probability $p$ of Erdős-Rényi graphs. Set $N=50$ with three base graphs: complete graph, star graph, and $k$-regular graph with $k=5$. Let the link connection probability $p$ vary from $p=0.1$ to $p=0.9$ at a stride of 0.1. We run Algorithms 1, 2, and 3, and plot the maximum regret $\mathsf{MReg}$ versus the probability $p$ in Figure 3 and Figure 4 for convex and strongly convex losses, respectively. From the figures, we observe the trend that the increasing of link probability $p$ can improve the algorithm performance.

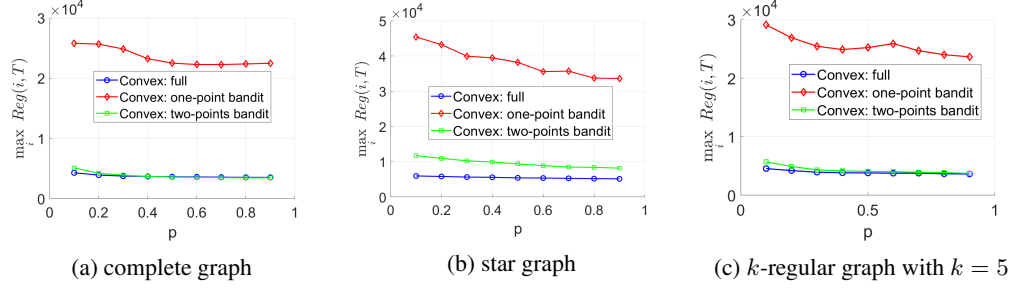

(a) complete graph        (b) star graph        (c) $k$-regular graph with $k = 5$

Figure 3: MReg vs probability for convex losses.

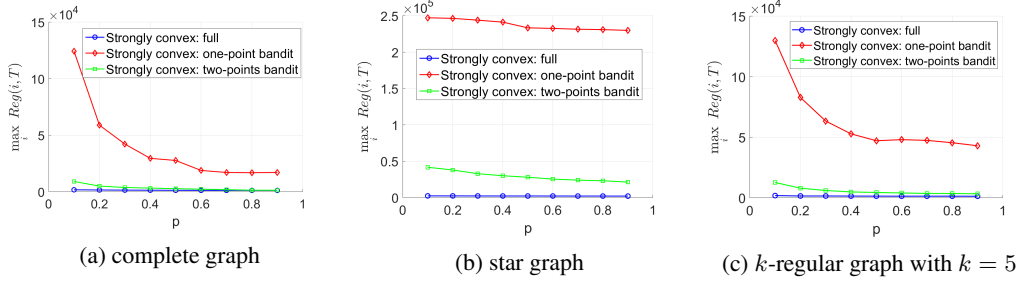

(a) complete graph        (b) star graph        (c) $k$-regular graph with $k = 5$

Figure 4: MReg vs probability for strongly convex losses.

Finally, we fix the time horizon $T = 200$ and let the base graph be $k$-regular graph with $k = 3$. We let the node number $N$ vary from $N = 5$ to $N = 80$, implement Algorithms 1, 2, and 3 over the **bodyfat** dataset, and plot $\frac{\max_j \mathsf{Reg}(j,T)}{N}$ versus $N$ in Figure 5 for convex and strongly convex losses. In addition, we set $N = 20$, let the vector dimension $d$ vary from $d = 5$ to $d = 100$, implement Algorithms 1, 2, and 3 on the randomly generated dataset, and plot $\frac{\max_j \mathsf{Reg}(j,T)}{d}$ versus $d$ in Figure 5 for convex and strongly convex losses.

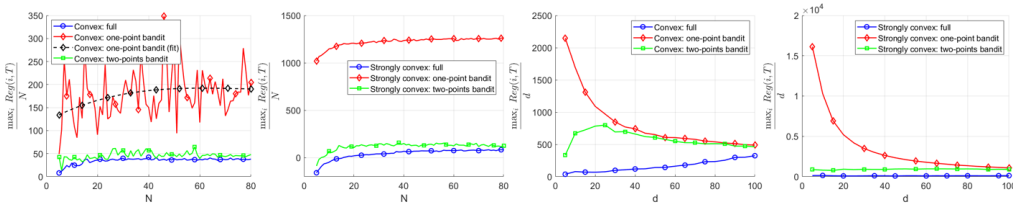

Figure 5: Algorithm performance vs the node number $N$ and the vector dimension $d$.

## 6 Conclusions

In this paper, we consider the online convex optimization over Erdős-Rényi random graphs under the full information feedback, one-point and two-points bandit feedback. We develop consensus-based distributed algorithms and establish the regret bounds for both convex and strongly convex losses, which match those of the centralized online optimization in the literature. We further quantitatively characterize the influence of the algebraic network connectivity on the regret bounds, and show that the link connection probability can be used to tune a trade-off between the communication overhead and the computation accuracy. Future directions include closing the gap of the regret bounds and extending the kernel-based methods (see [20]) to bandit online convex optimization over networks.

## Broader Impact Section

The work provides the theoretical understanding of the performance limits about distributed online convex optimization over random networks, and could be applied in processing streaming data to various Internet of Things systems, such as machine learning with personal wearable devices. Currently, it does not present any foreseeable societal consequence.

## Acknowledgment

The work was sponsored by Shanghai Sailing Program (No. 20YF1453000, No. 20YF1452800) and the Fundamental Research Funds for the Central Universities, China (No. 22120200047, No. 22120200048). The authors would like to thank Professor Deming Yuan for the helpful discussions, and thank Wenting Liu and Xiaoyu Ma for the help in numerical experiments.

## Footnotes

[2]The data set is from https://www.csie.ntu.edu.tw/ cjlin/libsvmtools/datasets/

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
