[Supplementary Material]

# Supplementary Material for "Online Convex Optimization Over Erdős-Rényi Random Networks"

**Jinlong Lei**
Tongji University
Shanghai, China
leijinlong@tongji.edu.cn

**Peng Yi**
Tongji University
Shanghai, China
yipeng@tongji.edu.cn

**Yiguang Hong**
Tongji University
Shanghai, China
yghong@iss.ac.cn

**Jie Chen**
Tongji University
Shanghai, China
chenjie@bit.edu.cn

**Guodong Shi,**
The University of Sydney
NSW, Australia
guodong.shi@sydney.edu.au

## A    Proofs of Section 2

### A.1    Preliminary Lemmas

In this subsection, we present some preliminary lemmas that will be used in the subsequent for proving the regret bounds. Without loss of generality, suppose that for each $i \in \mathcal{V}$ and $t = 1, \dots, T$, $f_{i,t}$ is $\alpha_t$-strongly convex with $\alpha_t \geq 0$, where $\alpha_t \equiv 0$ in the convex case. We start with a general lemma concerning the regret bound.

**Lemma 1.** *Let Assumptions 1 and 2 hold. Consider Algorithm 1, where $\{\eta_t\}$ is a non-increasing sequence.*
*(i) If $\alpha_t \equiv 0$, then for each $j \in \mathcal{V}$ :*

$$\mathsf{Reg}(j,T) \leq \frac{ND_1^2}{2\eta_T} + \frac{NG_f^2}{2} \sum_{t=1}^{T} \eta_t + G_f \sum_{t=1}^{T} \sum_{i=1}^{N} \|\mathbf{x}_{i,t} - \mathbf{x}_{j,t}\|. \tag{A.1}$$

*(ii) If $\alpha_t > 0$, by setting $\eta_t = \frac{1}{\sum_{\tau=1}^{t} \alpha_\tau}$ we obtain that for each $j \in \mathcal{V}$ :*

$$\mathsf{Reg}(j,T) \leq \frac{NG_f^2}{2} \sum_{t=1}^{T} \eta_t + G_f \sum_{t=1}^{T} \sum_{i=1}^{N} \|\mathbf{x}_{i,t} - \mathbf{x}_{j,t}\|. \tag{A.2}$$

**Proof.**    Define $a_{ij,t} \triangleq a$ if $\{i,j\} \in \mathrm{E}_t$, $a_{ii,t} \triangleq 1 - a|\mathrm{N}_{i,t}|$, and $a_{ij,t} = 0$, otherwise. Thus, $\sum_{j=1}^{N} a_{ij,t} = 1$ and $\sum_{i=1}^{N} a_{ij,t} = 1$. By using (3), $\mathbf{x}^* \in \mathcal{K}$, and the non-expansive property of the projection operator, we have that

$$\sum_{i=1}^{N} \|\mathbf{x}_{i,t+1} - \mathbf{x}^*\|^2 \leq \sum_{i=1}^{N} \left\| \sum_{j=1}^{N} a_{ij,t} \mathbf{y}_{j,t} - \mathbf{x}^* \right\|^2 \overset{(a)}{=} \sum_{i=1}^{N} \left\| \sum_{j=1}^{N} a_{ij,t} (\mathbf{y}_{j,t} - \mathbf{x}^*) \right\|^2$$

$$\overset{(b)}{\leq} \sum_{i=1}^{N} \sum_{j=1}^{N} a_{ij,t} \|\mathbf{y}_{j,t} - \mathbf{x}^*\|^2 \overset{(c)}{=} \sum_{j=1}^{N} \|\mathbf{y}_{j,t} - \mathbf{x}^*\|^2 \overset{(2)}{=} \sum_{i=1}^{N} \|\mathbf{x}_{i,t} - \mathbf{x}^* - \eta_t \nabla f_{i,t}(\mathbf{x}_{i,t})\|^2 \tag{A.3}$$

$$= \sum_{i=1}^{N} \|\mathbf{x}_{i,t} - \mathbf{x}^*\|^2 + \eta_t^2 \sum_{i=1}^{N} \|\nabla f_{i,t}(\mathbf{x}_{i,t})\|^2 - 2\eta_t \sum_{i=1}^{N} (\mathbf{x}_{i,t} - \mathbf{x}^*)^T \nabla f_{i,t}(\mathbf{x}_{i,t}),$$

where inequality (a) used $\sum_{j=1}^{N} a_{ij,t} = 1$, inequality (b) used the Jensen's inequality, and equality (c) used $\sum_{i=1}^{N} a_{ij,t} = 1$ for each $j \in \mathcal{V}$. It is noticed from Assumption 2 that

$$f_{i,t}(\mathbf{x}_{i,t}) = f_{i,t}(\mathbf{x}_{j,t}) + f_{i,t}(\mathbf{x}_{i,t}) - f_{i,t}(\mathbf{x}_{j,t})$$
$$\geq f_{i,t}(\mathbf{x}_{j,t}) + (\mathbf{x}_{i,t} - \mathbf{x}_{j,t})^T \nabla f_{i,t}(\mathbf{x}_{j,t}) \geq f_{i,t}(\mathbf{x}_{j,t}) - G_f \|\mathbf{x}_{i,t} - \mathbf{x}_{j,t}\|,$$

and hence

$$\sum_{i=1}^{N} \left( f_{i,t}(\mathbf{x}_{i,t}) - f_{i,t}(\mathbf{x}^*) \right) \geq \sum_{i=1}^{N} \left( f_{i,t}(\mathbf{x}_{j,t}) - f_{i,t}(\mathbf{x}^*) \right) - G_f \sum_{i=1}^{N} \|\mathbf{x}_{i,t} - \mathbf{x}_{j,t}\|. \tag{A.4}$$

Applying the definition of $\alpha_t$-strong convexity of $f_{i,t}$ to the pair of $\mathbf{x}_{i,t}, \mathbf{x}^*$, we obtain that

$$\left(\mathbf{x}_{i,t} - \mathbf{x}^*\right)^T \nabla f_{i,t}(\mathbf{x}_{i,t}) \geq \left( f_{i,t}(\mathbf{x}_{i,t}) - f_{i,t}(\mathbf{x}^*) \right) + \frac{\alpha_t}{2} \|\mathbf{x}_{i,t} - \mathbf{x}^*\|^2.$$

It combined with (A.4) produces

$$\sum_{i=1}^{N} \left(\mathbf{x}_{i,t} - \mathbf{x}^*\right)^T \nabla f_{i,t}(\mathbf{x}_{i,t})$$
$$\geq \sum_{i=1}^{N} \left( f_{i,t}(\mathbf{x}_{j,t}) - f_{i,t}(\mathbf{x}^*) \right) - G_f \sum_{i=1}^{N} \|\mathbf{x}_{i,t} - \mathbf{x}_{j,t}\| + \frac{\alpha_t}{2} \sum_{i=1}^{N} \|\mathbf{x}_{i,t} - \mathbf{x}^*\|^2. \tag{A.5}$$

By substituting (A.5) into (A.3) and using Assumption 2, we derive

$$\sum_{i=1}^{N} \|\mathbf{x}_{i,t+1} - \mathbf{x}^*\|^2 \leq \sum_{i=1}^{N} \|\mathbf{x}_{i,t} - \mathbf{x}^*\|^2 + N\eta_t^2 G_f^2 - 2\eta_t \sum_{i=1}^{N} \left( f_{i,t}(\mathbf{x}_{j,t}) - f_{i,t}(\mathbf{x}^*) \right)$$
$$+ 2G_f \eta_t \sum_{i=1}^{N} \|\mathbf{x}_{i,t} - \mathbf{x}_{j,t}\| - \alpha_t \eta_t \sum_{i=1}^{N} \|\mathbf{x}_{i,t} - \mathbf{x}^*\|^2. \tag{A.6}$$

By rearranging the terms, there holds

$$\sum_{i=1}^{N} \left( f_{i,t}(\mathbf{x}_{j,t}) - f_{i,t}(\mathbf{x}^*) \right) \leq \frac{(1 - \alpha_t \eta_t) \sum_{i=1}^{N} \|\mathbf{x}_{i,t} - \mathbf{x}^*\|^2 - \sum_{i=1}^{N} \|\mathbf{x}_{i,t+1} - \mathbf{x}^*\|^2}{2\eta_t}$$
$$+ NG_f^2 \eta_t / 2 + G_f \sum_{i=1}^{N} \|\mathbf{x}_{i,t} - \mathbf{x}_{j,t}\|.$$

By summing up the above inequality from $t = 1$ to $T$, we obtain that

$$\sum_{t=1}^{T} \sum_{i=1}^{N} \left( f_{i,t}(\mathbf{x}_{j,t}) - f_{i,t}(\mathbf{x}^*) \right) \leq \frac{1}{2} \sum_{t=1}^{T} \sum_{i=1}^{N} \|\mathbf{x}_{i,t} - \mathbf{x}^*\|^2 \left( \frac{1}{\eta_t} - \frac{1}{\eta_{t-1}} - \alpha_t \right)$$
$$+ \frac{NG_f^2}{2} \sum_{t=1}^{T} \eta_t + G_f \sum_{t=1}^{T} \sum_{i=1}^{N} \|\mathbf{x}_{i,t} - \mathbf{x}_{j,t}\|, \quad \frac{1}{\eta_0} \triangleq 0. \tag{A.7}$$

(i) By using Assumption 1 and the non-increasing of $\{\eta_t\}$, we obtained that

$$\sum_{t=1}^{T} \sum_{i=1}^{N} \|\mathbf{x}_{i,t} - \mathbf{x}^*\|^2 \left( \frac{1}{\eta_t} - \frac{1}{\eta_{t-1}} \right) \leq \sum_{t=1}^{T} \sum_{i=1}^{N} D_1^2 \left( \frac{1}{\eta_t} - \frac{1}{\eta_{t-1}} \right) = \frac{ND_1^2}{\eta_T}.$$

This combined with (A.7) and $\alpha_t \equiv 0$ proves the bound (A.1).

(ii) From $\eta_t = \frac{1}{\sum_{\tau=1}^{t} \alpha_\tau}$ it follows that $\frac{1}{\eta_t} - \frac{1}{\eta_{t-1}} - \alpha_t = 0$. Hence by (A.7), we obtain (A.2). $\quad\square$

Let $\mathbf{I}_N$ denote the $N \times N$ identity matrix. Denote by $\mathbf{L}_t$ the Laplacian matrix of the graph $\mathrm{G}_t$, where $[\mathbf{L}_t]_{ij} = -1$ if $\{i, j\} \in \mathrm{E}_t$, $[\mathbf{L}_t]_{ii} = |\mathrm{N}_{i,t}|$, and and $[\mathbf{L}_t]_{ij} = 0$, otherwise. Then based on the Erdös-Rényi rule that $\{i, j\} \in \mathrm{E}_t$ with probability $0 < p < 1$ for all $\{i, j\} \in \mathcal{E}$, we have

that $\mathbb{E}[\mathbf{L}_t]_{ij} = -p$ if $\{i, j\} \in \mathcal{E}$, $\mathbb{E}[\mathbf{L}_t]_{ii} = p|\mathcal{N}_i|$, and and $\mathbb{E}[\mathbf{L}_t]_{ij} = 0$, otherwise. Therefore, $\mathbb{E}[\mathbf{L}_t] = p\mathbf{L}$. We further define $\mathbf{A}_t \triangleq \mathbf{I}_N - a\mathbf{L}_t$,

$$\mathbf{\Phi}(t, t+1) \triangleq \mathbf{I}_N \text{ and } \mathbf{\Phi}(t, s) \triangleq \mathbf{A}_t \cdots \mathbf{A}_s, \ \forall t \geq s \geq 1. \tag{A.8}$$

By the definition of $\mathbf{A}_t$ it is seen that $\mathbf{A}_t$ is a positive and symmetric matrix with the sum of each row equal to 1. Then for any $t \geq 1$:

$$\mathbb{E}[\mathbf{A}_t] \triangleq \bar{\mathbf{A}} = \mathbf{I}_N - ap\mathbf{L},$$
$$\mathbb{E}[\mathbf{A}_t^2] = \mathbf{I}_N - 2ap\mathbf{L} + a^2 \mathbb{E}[\mathbf{L}_t^2].$$

Let $\bar{\mathcal{G}} = \{\mathcal{V}, \bar{\mathcal{E}}\}$ be an undirected graph generated by the matrix $\mathbb{E}[\mathbf{A}_t^2]$, where $\{i, j\} \in \bar{\mathcal{E}}$ if $(i, j)_{th}$ entry of $\mathbb{E}[\mathbf{A}_t^2]$ satisfies $\mathbb{E}[\mathbf{A}_t^2]_{ij} > 0$. Note by $0 < a \leq \frac{1}{1 + \max_i |\mathcal{N}_i|}$ and $0 < p < 1$ that for each pair $\{i, j\} \in \mathcal{E}$:

$$\mathbb{E}[\mathbf{A}_t^2]_{ij} \geq \mathbb{E}[a_{ii,t} a_{ij,t} + a_{ij,t} a_{jj,t}] = ap\left(2 - ap|\mathcal{N}_i| - ap|\mathcal{N}_j|\right) > 0.$$

Hence, $\{i, j\} \in \bar{\mathcal{E}}$ if $\{i, j\} \in \mathcal{E}$. By the fact that the base graph $\mathcal{G}$ is connected, $\bar{\mathcal{G}}$ is also an undirected and connected graph. We can similarly show that the graph associated with the matrix $\bar{\mathbf{A}}$ is undirected and connected. Then we obtain the following with $\mathbf{\Omega} \triangleq \frac{\mathbf{1}_N \mathbf{1}_N^T}{N}$:

$$\rho_0 = \|\bar{\mathbf{A}} - \mathbf{\Omega}\| = \text{esp}(\bar{\mathbf{A}}) = \max\{|\lambda| : \lambda \text{ is the eigenvalue of } \bar{\mathbf{A}} \text{ different from 1}\},$$
$$\rho^2 = \|\mathbb{E}[\mathbf{A}_t^2] - \mathbf{\Omega}\| = \text{esp}\left(\mathbf{I}_N - 2ap\mathbf{L} + a^2 \mathbb{E}[\mathbf{L}_t^2]\right). \tag{A.9}$$

Next, we establish a lower bound and an upper bound on the consensus matrix, which is important for estimating the consensus error.

**Lemma 2.** *Define* $\mathcal{F}_s \triangleq \sigma\{\mathbf{e}_1, \mathbf{A}_1, \cdots, \mathbf{A}_{s-1}\}$ *for any* $s \geq 1$. *Let* $\mathbf{e}_{t+1} \triangleq \left(\mathbf{\Phi}(t, s) - \mathbf{\Omega}\right)\mathbf{e}_s$ *for any nonzero vector* $\mathbf{e}_s \in \mathbb{R}^N$ *adapted to* $\mathcal{F}_s$. *Then the following holds:*

$$\rho_0^{t-s+1} \leq \max_{\mathbf{e}_s \in \mathbb{R}^N} \frac{\mathbb{E}\left[\|\mathbf{e}_{t+1}\||\mathcal{F}_s\right]}{\|\mathbf{e}_s\|} \leq \rho^{t-s+1}. \tag{A.10}$$

**Proof.** Since $\mathbf{A}_t \mathbf{\Omega} = \mathbf{\Omega}$, by the definition of $\mathbf{\Phi}(t, s)$, we obtain that

$$\left(\mathbf{A}_t - \mathbf{\Omega}\right) \cdots \left(\mathbf{A}_s - \mathbf{\Omega}\right) = \mathbf{\Phi}(t, s) - \mathbf{\Omega}, \quad \forall t \geq s \geq 1.$$

Note that $\mathbf{A}_t$ is independent of $\mathcal{F}_t = \sigma\{\mathbf{e}_1, \mathbf{A}_1, \cdots, \mathbf{A}_{t-1}\}$. Hence for any $t \geq s \geq 1$:

$$\mathbb{E}\left[\mathbf{\Phi}(t, s)|\mathcal{F}_s\right] = \mathbb{E}\left[\mathbb{E}\left[\mathbf{\Phi}(t, s)|\mathcal{F}_t\right]\Big|\mathcal{F}_s\right]$$
$$= \mathbb{E}\left[\mathbb{E}\left[\left(\mathbf{A}_t - \mathbf{\Omega}\right)\mathbf{\Phi}(t-1, s)|\mathcal{F}_t\right]\Big|\mathcal{F}_s\right] = (\bar{\mathbf{A}} - \mathbf{\Omega})\mathbb{E}\left[\mathbf{\Phi}(t-1, s)|\mathcal{F}_s\right],$$

where the first equality holds by [1, Chapter 7, Eqn. (14v)] because $\mathcal{F}_s \subset \mathcal{F}_t$. Then based on the above recursion and $\bar{\mathbf{A}}\mathbf{\Omega} = \mathbf{\Omega}$, we obtain that $\mathbb{E}\left[\mathbf{\Phi}(t, s)|\mathcal{F}_s\right] = \bar{\mathbf{A}}^{t-s+1} - \mathbf{\Omega}$. Then by the fact that $\mathbf{e}_s$ is adapted to $\mathcal{F}_s$, the following holds for any $t \geq s \geq 1$:

$$\mathbb{E}\left[\mathbf{e}_{t+1}|\mathcal{F}_s\right] = \mathbb{E}\left[\left(\mathbf{\Phi}(t, s) - \mathbf{\Omega}\right)\mathbf{e}_s|\mathcal{F}_s\right] = (\bar{\mathbf{A}}^{t-s+1} - \mathbf{\Omega})\mathbf{e}_s.$$

Then by the Jensen's inequality for conditional expectations, the following holds

$$\mathbb{E}\left[\|\mathbf{e}_{t+1}\||\mathcal{F}_s\right] \geq \left\|\mathbb{E}[\mathbf{e}_{t+1}|\mathcal{F}_s]\right\| = \left\|(\bar{\mathbf{A}}^{t-s+1} - \mathbf{\Omega})\mathbf{e}_s\right\|, \quad \forall t \geq s \geq 1. \tag{A.11}$$

Note that $\mathbf{A}_t \mathbf{\Omega} = \mathbf{A}_t^T \mathbf{\Omega} = \mathbf{\Omega}$ and $\mathbf{A}_t^T \mathbf{A}_t = \mathbf{A}_t^2$. Then for any $t \geq s \geq 1$:

$$\mathbb{E}\left[\mathbf{e}_{t+1}^T \mathbf{e}_{t+1}|\mathcal{F}_s\right] = \mathbb{E}\left[\mathbb{E}\left[\mathbf{e}_{t+1}^T \mathbf{e}_{t+1}|\mathcal{F}_t\right]\Big|\mathcal{F}_s\right]$$
$$= \mathbb{E}\left[\mathbb{E}\left[\mathbf{e}_t^T (\mathbf{A}_t - \mathbf{\Omega})^T (\mathbf{A}_t - \mathbf{\Omega})\mathbf{e}_t|\mathcal{F}_t\right]\Big|\mathcal{F}_s\right] = \mathbb{E}\left[\mathbf{e}_t^T \mathbb{E}[\mathbf{A}_t^2 - \mathbf{\Omega}]\mathbf{e}_t\Big|\mathcal{F}_s\right]$$
$$\leq \mathbb{E}\left[\mathbf{e}_t^T \mathbf{e}_t|\mathcal{F}_s\right]\left\|\mathbb{E}[\mathbf{A}_1^2] - \mathbf{\Omega}\right\| \leq \ldots \leq \mathbf{e}_s^T \mathbf{e}_s \left\|\mathbb{E}[\mathbf{A}_1^2] - \mathbf{\Omega}\right\|^{t-s+1},$$

where the third equality holds because $\mathbf{e}_t$ is adapted to $\mathcal{F}_t$ and $\mathbf{A}_t$ is independent of $\mathcal{F}_t$. Then by the Jensen's inequality for conditional expectations, we obtain that

$$\mathbb{E}\big[\|\mathbf{e}_{t+1}\|\,|\,\mathcal{F}_s\big] \leq \sqrt{\mathbb{E}\big[\mathbf{e}_{t+1}^T\mathbf{e}_{t+1}\,|\,\mathcal{F}_s\big]} \leq \sqrt{\mathbf{e}_s^T\mathbf{e}_s}\,\big\|\mathbb{E}[\mathbf{A}_1^2] - \mathbf{\Omega}\big\|^{(t-s+1)/2}. \tag{A.12}$$

Therefore, by combing (A.11) with (A.12), we obtain that for any $t \geq s \geq 1$:

$$\left\|(\bar{\mathbf{A}}^{t-s+1} - \mathbf{\Omega})\frac{\mathbf{e}_s}{\|\mathbf{e}_s\|}\right\| \leq \frac{\mathbb{E}\big[\|\mathbf{e}_{t+1}\|\,|\,\mathcal{F}_s\big]}{\|\mathbf{e}_s\|} \leq \big\|\mathbb{E}[\mathbf{A}_1^2] - \mathbf{\Omega}\big\|^{(t-s+1)/2}$$

Thus, by maximizing the above equation with respect to $\mathbf{e}_s$, using (A.9) and recalling the definition of the matrix two-norm $\|\mathbf{A}\| = \max\limits_{\mathbf{x} \ s.t. \ \|\mathbf{x}\|=1}\|\mathbf{A}\mathbf{x}\|$, we proves (A.10). $\qquad\square$

**Remark 1.** *The upper bound established in Lemma 2 might be obtained by some specific selection of Erdős-Rényi random graphs. For example [2, Example 4.7], the priori graph $\mathcal{G} = \{\mathcal{V},\mathcal{E}\}$ is a complete graph and $a = \frac{1}{N}$.*

Then based on Lemma 2, we can establish the following lemma concerning the consensus error.

**Lemma 3.** *Suppose Assumptions 1, and 2, hold. Let the local estimates $\{\mathbf{x}_{i,t}\}_{t=1}^T$ for each node $i \in \mathcal{V}$ be generated by Algorithm 1. Then the following hold with $\bar{\mathbf{x}}_t = \frac{1}{N}\sum_{i=1}^N\mathbf{x}_{i,t}$:*

$$\sum_{i=1}^N \mathbb{E}\big[\|\mathbf{x}_{i,t} - \bar{\mathbf{x}}_t\|\big] \leq 3NG_f\sum_{s=1}^{t-1}\eta_s\rho^{t-s}, \text{ and}$$

$$\max_{j\in\mathcal{V}}\mathbb{E}\big[\|\mathbf{x}_{i,t} - \bar{\mathbf{x}}_t\|\big] \leq 3\sqrt{N}G_f\sum_{s=1}^{t-1}\eta_s\rho^{t-s}. \tag{A.13}$$

**Proof.** Note by (3) and the definition of $a_{ij,t}$ that $\mathbf{x}_{i,t+1} = \Pi_\mathcal{K}\left(\sum_{j=1}^N a_{ij,t}\mathbf{y}_{j,t}\right)$. Define

$$\mathbf{r}_{i,t+1} = \mathbf{x}_{i,t+1} - \sum_{j=1}^N a_{ij,t}\mathbf{y}_{j,t} = \Pi_\mathcal{K}\left(\sum_{j=1}^N a_{ij,t}\mathbf{y}_{j,t}\right) - \sum_{j=1}^N a_{ij,t}\mathbf{y}_{j,t}. \tag{A.14}$$

Then by substituting (2) into (A.14), we obtain that

$$\|\mathbf{r}_{i,t+1}\| = \left\|\Pi_\mathcal{K}\left(\sum_{j=1}^N a_{ij,t}\left(\mathbf{x}_{j,t} - \eta_t\nabla f_{j,t}(\mathbf{x}_{j,t})\right)\right) - \sum_{j=1}^N a_{ij,t}\left(\mathbf{x}_{j,t} - \eta_t\nabla f_{j,t}(\mathbf{x}_{j,t})\right)\right\|$$

$$\overset{(a)}{\leq} \left\|\Pi_\mathcal{K}\left(\sum_{j=1}^N a_{ij,t}\left(\mathbf{x}_{j,t} - \eta_t\nabla f_{j,t}(\mathbf{x}_{j,t})\right)\right) - \sum_{j=1}^N a_{ij,t}\mathbf{x}_{j,t}\right\| + \eta_t\left\|\sum_{j=1}^N a_{ij,t}\nabla f_{j,t}(\mathbf{x}_{j,t})\right\|$$

$$\overset{(b)}{=} 2\eta_t\sum_{j=1}^N a_{ij,t}\|\nabla f_{j,t}(\mathbf{x}_{j,t})\| \overset{(c)}{\leq} 2\eta_t G_f, \ \forall i \in \mathcal{V}, \tag{A.15}$$

where (a) used the triangle inequality, (b) used the non-expansive property of the projection operator and the fact that $\sum_{j=1}^N a_{ij,t}\mathbf{x}_{j,t} \in \mathcal{K}$ by $\sum_{j=1}^N a_{ij,t} = 1$, and (c) holds by Assumption 2 and $\sum_{j=1}^N a_{ij,t} = 1$. By combing (2) with (A.14) and (A.8), there holds

$$\mathbf{x}_{i,t+1} = \sum_{j=1}^N a_{ij,t}\mathbf{y}_{j,t} + \mathbf{r}_{i,t+1} = \sum_{j=1}^N a_{ij,t}\left(\mathbf{x}_{i,t} - \eta_t\nabla f_{i,t}(\mathbf{x}_{i,t})\right) + \mathbf{r}_{i,t+1}.$$

Then by stacking the above equation for each $i \in \mathcal{V}$, and using $\mathbf{x}_{i,1} = \mathbf{0}$ for each $i \in \mathcal{V}$, there holds

$$\mathbf{x}_{t+1} \triangleq \begin{pmatrix} \mathbf{x}_{1,t+1} \\ \vdots \\ \mathbf{x}_{N,t+1} \end{pmatrix} = \mathbf{A}_t \otimes \mathbf{I}_d\left(\mathbf{x}_{t+1} - \eta_t\begin{pmatrix} \nabla f_{1,s}(\mathbf{x}_{1,t}) \\ \vdots \\ \nabla f_{N,s}(\mathbf{x}_{N,t}) \end{pmatrix}\right) + \begin{pmatrix} \mathbf{r}_{1,t+1} \\ \vdots \\ \mathbf{r}_{N,t+1} \end{pmatrix}$$

$$\overset{(A.8)}{=} -\sum_{s=1}^t \eta_s\mathbf{\Phi}(t,s) \otimes \mathbf{I}_d\begin{pmatrix} \nabla f_{1,s}(\mathbf{x}_{1,s}) \\ \vdots \\ \nabla f_{N,s}(\mathbf{x}_{N,s}) \end{pmatrix} + \sum_{s=1}^t \mathbf{\Phi}(t,s) \otimes \mathbf{I}_d\begin{pmatrix} \mathbf{r}_{1,s+1} \\ \vdots \\ \mathbf{r}_{N,s+1} \end{pmatrix}.$$

Thus by the definition of $\bar{\mathbf{x}}_t$, and using the doubly stochastic of $\mathbf{\Phi}(t,s)$, we obtain that

$$\bar{\mathbf{x}}_{t+1} = \frac{1}{N}\sum_{i=1}^N \mathbf{x}_{i,t+1} = -\sum_{s=1}^t \eta_s \frac{1}{N}\sum_{j=1}^N \nabla f_{j,s}(\mathbf{x}_{j,s}) + \sum_{s=1}^t \frac{1}{N}\sum_{j=1}^N \mathbf{r}_{j,s+1}.$$

Then we obtain the following

$$\widetilde{\mathbf{x}}_{t+1} \triangleq \begin{pmatrix} \mathbf{x}_{1,t+1} - \bar{\mathbf{x}}_{t+1} \\ \vdots \\ \mathbf{x}_{N,t+1} - \bar{\mathbf{x}}_{t+1} \end{pmatrix} = -\sum_{s=1}^t \eta_s (\mathbf{\Phi}(t,s) - \mathbf{\Omega}) \otimes \mathbf{I}_d \begin{pmatrix} \nabla f_{1,s}(\mathbf{x}_{1,s}) \\ \vdots \\ \nabla f_{N,s}(\mathbf{x}_{N,s}) \end{pmatrix}$$

$$+ \sum_{s=1}^t (\mathbf{\Phi}(t,s) - \mathbf{\Omega}) \otimes \mathbf{I}_d \begin{pmatrix} \mathbf{r}_{1,s+1} \\ \vdots \\ \mathbf{r}_{N,s+1} \end{pmatrix}.$$

Thus, from (A.10), (A.15), and Assumption 2 it follows that

$$\mathbb{E}\big[\,\|\widetilde{\mathbf{x}}_{t+1}\|\,\big|\mathcal{F}_s\big] \leq \sum_{s=1}^t \rho^{t-s+1}\left(\eta_s\left\|\begin{pmatrix} \nabla f_{1,s}(\mathbf{x}_{1,s}) \\ \vdots \\ \nabla f_{N,s}(\mathbf{x}_{N,s}) \end{pmatrix}\right\| + \left\|\begin{pmatrix} \mathbf{r}_{1,s+1} \\ \vdots \\ \mathbf{r}_{N,s+1} \end{pmatrix}\right\|\right)$$

$$\leq 3\sqrt{N}G_f \sum_{s=1}^t \eta_s \rho^{t-s+1}.$$

By taking unconditional expectation with respect to the above equation, there holds

$$\mathbb{E}\big[\,\|\widetilde{\mathbf{x}}_{t+1}\|\,\big] \leq 3\sqrt{N}G_f \sum_{s=1}^t \eta_s \rho^{t-s+1}. \tag{A.16}$$

Thus, $\mathbb{E}\,[\|\mathbf{x}_{j,t} - \bar{\mathbf{x}}_t\|] \leq 3\sqrt{N}G_f \sum_{s=1}^{t-1} \eta_s \rho^{t-s}$ for each $j \in \mathcal{V}$. Note by the Jensen's inequality that $\left(\sum_{i=1}^N x_i/N\right)^2 \leq \sum_{i=1}^N x_i^2/N$, which implies that $\sum_{i=1}^N x_i \leq \sqrt{N\sum_{i=1}^N x_i^2}$. This incorporating with (A.16) produces

$$\mathbb{E}\left[\sum_{i=1}^N \|\mathbf{x}_{i,t} - \bar{\mathbf{x}}_t\|\right] \leq \mathbb{E}\left[\sqrt{N\sum_{i=1}^N \|\mathbf{x}_{i,t} - \bar{\mathbf{x}}_t\|^2}\right] = \sqrt{N}\mathbb{E}\big[\,\|\widetilde{\mathbf{x}}_t\|\,\big] \leq 3NG_f \sum_{s=1}^{t-1} \eta_s \rho^{t-s}.$$

Thus, the lemma is proved. $\qquad\square$

## A.2  Proof of Theorem 1

Note that

$$\sum_{i=1}^N \|\mathbf{x}_{i,t} - \mathbf{x}_{j,t}\| = \sum_{i=1}^N \|\mathbf{x}_{i,t} - \bar{\mathbf{x}}_t - (\mathbf{x}_{j,t} - \bar{\mathbf{x}}_t)\| \leq \sum_{i=1}^N \|\mathbf{x}_{i,t} - \bar{\mathbf{x}}_t\| + N\|\mathbf{x}_{j,t} - \bar{\mathbf{x}}_t\|.$$

Then from (A.13) it follows that

$$\mathbb{E}\left[\sum_{i=1}^N \|\mathbf{x}_{i,t} - \mathbf{x}_{j,t}\|\right] \leq \sum_{i=1}^N \mathbb{E}\,[\|\mathbf{x}_{i,t} - \bar{\mathbf{x}}_t\|] + N\mathbb{E}\,[\|\mathbf{x}_{j,t} - \bar{\mathbf{x}}_t\|]$$

$$\leq (3N + 3N^{3/2})G_f \sum_{s=1}^{t-1} \eta_s \rho^{t-s}. \tag{A.17}$$

It is noticed that

$$\sum_{t=1}^T \sum_{s=1}^{t-1} \eta_s \rho^{t-s} = \sum_{t=1}^{T-1} \sum_{s=t+1}^T \eta_{t-s} \rho^t = \sum_{t=1}^{T-1} \rho^t \sum_{s=t+1}^T \eta_{t-s} \leq \frac{\rho}{1-\rho}\sum_{s=1}^T \eta_s = \frac{\rho}{1-\rho}\sum_{t=1}^T \eta_t.$$

This combined with (A.17) produces

$$\sum_{t=1}^{T}\mathbb{E}\left[\sum_{i=1}^{N}\|\mathbf{x}_{i,t}-\mathbf{x}_{j,t}\|\right]\leq\frac{\rho(3N+3N^{3/2})G_f}{1-\rho}\sum_{s=1}^{T}\eta_s. \tag{A.18}$$

Note that $\sum_{t=1}^{T}\frac{1}{\sqrt{t}}\leq\int_{0}^{T}\frac{1}{\sqrt{x}}dx=2\sqrt{x}|_{0}^{T}=2\sqrt{T}$. Then by recalling that $\eta_t=\frac{D_1}{G_f\sqrt{t}}$, taking the unconditional expectation on both sides of (A.1) and using (A.18), we obtain that

$$\mathbb{E}\big[Reg(j,T)\big]\leq\frac{ND_1G_f\sqrt{T}}{2}+ND_1G_f\sqrt{T}+\frac{6\rho N(1+\sqrt{N})D_1G_f\sqrt{T}}{1-\rho}.$$

Then the theorem is proved. □

### A.3    Proof of Theorem 2

By taking the unconditional expectation on both sides of (A.2) and using (A.18), we obtain

$$\mathbb{E}\big[Reg(j,T)\big]\leq\frac{NG_f^2}{2}\left(1+\frac{6\rho(1+\sqrt{N})}{1-\rho}\right)\sum_{t=1}^{T}\eta_t. \tag{A.19}$$

Note from $\eta_t=\frac{1}{\alpha t}$ that

$$\sum_{t=1}^{T}\eta_t=\frac{1}{\alpha}+\frac{1}{\alpha}\sum_{t=2}^{T}\frac{1}{t}\leq\frac{1}{\alpha}+\frac{1}{\alpha}\int_{1}^{T}\frac{1}{x}dx=\frac{1}{\alpha}+\frac{1}{\alpha}\ln(x)|_{1}^{T}=\frac{1}{\alpha}(1+\ln(T)).$$

This combined with (A.19) proves the theorem. □

## B    Proofs of Section 3

*Proof of Theorem 3.* By Assumption 4 and $\xi=\delta/r$ that for any $\mathbf{x}\in(1-\xi)\mathcal{K}:\mathbf{x}+\delta\mathbf{u}\subseteq(1-\xi)\mathcal{K}+\xi r\mathcal{B}\subseteq(1-\xi)\mathcal{K}+\xi\mathcal{K}\subseteq\mathcal{K}$. Then from (6) and (8) it follows that for each $i\in\mathcal{V}$:

$$\|\mathbf{g}_{i,t}\|\leq\frac{d}{\delta}\|f_{i,t}(\mathbf{x}_{i,t}+\delta\mathbf{u}_{i,t})\|\|\mathbf{u}_{i,t}\|\leq\frac{dC}{\delta},\quad t=1,\cdots,T. \tag{B.1}$$

Then by $\nabla\hat{f}_{i,t}(\mathbf{x}_{i,t})=\mathbb{E}[\mathbf{g}_{i,t}]$, $\|\nabla\hat{f}_{i,t}(\mathbf{x}_{i,t})\|\leq\frac{dC}{\delta}\triangleq G_f$ holds for each $i\in\mathcal{V}$ and any $t=1,\cdots,T$. Note by Assumption 4 that $\|\mathbf{x}-\mathbf{y}\|\leq 2R\triangleq D_1$ for any $\mathbf{x},\mathbf{y}\in(1-\xi)\mathcal{K}$. By recalling the definition (1), similarly to Theorem 1, we can show that for each $j\in\mathcal{V}$:

$$\mathbb{E}\Big[\sum_{t=1}^{T}\sum_{i=1}^{N}\hat{f}_{i,t}(\mathbf{x}_{j,t})\Big]-\min_{\mathbf{x}\in(1-\xi)\mathcal{K}}\sum_{t=1}^{T}\sum_{i=1}^{N}\hat{f}_{i,t}(\mathbf{x})\leq\frac{3dNRC}{\delta}\left(1+\frac{4\rho(1+\sqrt{N})}{1-\rho}\right)\sqrt{T}. \tag{B.2}$$

Since $\mathbf{x}\in(1-\xi)\mathcal{K}\subseteq\mathcal{K}$ and $\mathbf{x}+\delta\mathbf{u}\in\mathcal{K}$, by Assumption 5 and the definition of $\hat{f}_{i,t}$ that

$$\begin{aligned}\|\hat{f}_{i,t}(\mathbf{x})-f_{i,t}(\mathbf{x})\|&=\|\mathbb{E}_{\mathbf{u}\in\mathcal{B}}[f_{i,t}(\mathbf{x}+\delta\mathbf{u})]-f_{i,t}(\mathbf{x})\|\\&\leq\mathbb{E}_{\mathbf{u}\in\mathcal{B}}\|f_{i,t}(\mathbf{x}+\delta\mathbf{u})-f_{i,t}(\mathbf{x})\|\leq\delta L_f,\quad\forall\mathbf{x}\in(1-\xi)\mathcal{K}.\end{aligned}$$

Therefore, we obtain that $\hat{f}_{i,t}(\mathbf{x}_{j,t})\geq f_{i,t}(\mathbf{x}_{j,t})-\delta L_f$ and $\hat{f}_{i,t}(\mathbf{x})\leq f_{i,t}(\mathbf{x})+\delta L_f$. This combined with (B.2) produces

$$\begin{aligned}\mathbb{E}\Big[\sum_{t=1}^{T}\sum_{i=1}^{N}f_{i,t}(\mathbf{x}_{j,t})-\delta L_f\Big]&-\min_{\mathbf{x}\in(1-\xi)\mathcal{K}}\sum_{t=1}^{T}\sum_{i=1}^{N}\big(f_{i,t}(\mathbf{x})+\delta L_f\big)\\&\leq\frac{3dNRC}{\delta}\left(1+\frac{4\rho(1+\sqrt{N})}{1-\rho}\right)\sqrt{T}.\end{aligned}$$

By rearranging the terms, we obtain that

$$\mathbb{E}\Big[\sum_{t=1}^{T}\sum_{i=1}^{N}f_{i,t}(\mathbf{x}_{j,t})\Big] - \min_{\mathbf{x}\in(1-\xi)\mathcal{K}}\sum_{t=1}^{T}\sum_{i=1}^{N}f_{i,t}(\mathbf{x})$$

$$\leq \frac{3dNRC}{\delta}\left(1+\frac{4\rho(1+\sqrt{N})}{1-\rho}\right)\sqrt{T}+2\delta NL_f T.$$

Note by [3, Observation 1] that

$$\min_{\mathbf{x}\in(1-\xi)\mathcal{K}}\sum_{t=1}^{T}\sum_{i=1}^{N}f_{i,t}(\mathbf{x}) \leq 2\xi CTN + \min_{\mathbf{x}\in\mathcal{K}}\sum_{t=1}^{T}\sum_{i=1}^{N}f_{i,t}(\mathbf{x}). \tag{B.3}$$

Hence by the definition (1) and $\xi=\delta/r$, there holds

$$\mathbb{E}\big[Reg(j,T)\big] \leq \frac{3NdRC}{\delta}\left(1+\frac{4\rho(1+\sqrt{N})}{1-\rho}\right)\sqrt{T}+2\delta NL_f T + 2\delta CTN/r.$$

Hence, by the definitions of $c_1$ and $c_2$ that $\mathbb{E}\big[Reg(j,T)\big] \leq N\left(\frac{c_1\sqrt{T}}{\delta}+c_2\delta T\right)$. Thus, we complete the proof by using $\delta=(c_1/c_2)^{0.5}T^{-0.25}$. $\qquad\square$

*Proof of Theorem 4.* Recall by (B.1) that $G_f=\frac{dC}{\delta}$. We can obtain from Theorem 2 and the definition (1) that for each $j\in\mathcal{V}$:

$$\mathbb{E}\Big[\sum_{t=1}^{T}\sum_{i=1}^{N}\hat{f}_{i,t}(\mathbf{x}_{j,t})\Big] - \min_{\mathbf{x}\in(1-\xi)\mathcal{K}}\sum_{t=1}^{T}\sum_{i=1}^{N}\hat{f}_{i,t}(\mathbf{x}) \leq \frac{Nd^2C^2}{2\alpha\delta^2}\left(1+\frac{6\rho(1+\sqrt{N})}{1-\rho}\right)(1+\ln(T)).$$

Then by taking a similar procedure as the proof of Theorem 3 after (B.2), we have that

$$\mathbb{E}\big[Reg(j,T)\big] \leq \frac{Nd^2C^2}{2\alpha\delta^2}\left(1+\frac{6\rho(1+\sqrt{N})}{1-\rho}\right)(1+\ln(T))+2\delta NL_f T + 2\delta CTN/r$$

$$= N\left(\frac{c_3}{\delta^2}(1+\ln(T))+c_2\delta T\right).$$

Then we obtain the result by the definitions of $c_2, c_3$ and $\delta$. $\qquad\square$

## C  Proofs of Section 4

*Proof of Theorem 5.* By recalling that $\mathbf{x}\in(1-\xi)\mathcal{K}\subseteq\mathcal{K}$ and $\mathbf{x}+\delta\mathbf{u}\in\mathcal{K}$, from (9) and Assumption 5 that for each $i\in\mathcal{V}$ and any $t=1,\cdots,T$:

$$\|\tilde{\mathbf{g}}_{i,t}\| \leq \frac{d}{2\delta}\|f_{i,t}(\mathbf{x}_{i,t}+\delta\mathbf{u}_{i,t})-f_{i,t}(\mathbf{x}_{i,t}-\delta\mathbf{u}_{i,t})\|\|\mathbf{u}_{i,t}\| \leq \frac{d}{2\delta}2L_f\delta\|\mathbf{u}_{i,t}\|^2 \leq dL_f.$$

Then by $\nabla\hat{f}_{i,t}(\mathbf{x}_{i,t})=\mathbb{E}[\mathbf{g}_{i,t}]$, $\|\nabla\hat{f}_{i,t}(\mathbf{x}_{i,t})\| \leq dL_f \triangleq G_f$. Note by Assumption 4 that for any $\mathbf{x},\mathbf{y}\in(1-\xi)\mathcal{K}: \|\mathbf{x}-\mathbf{y}\| \leq 2R \triangleq D_1$. We then obtain from Theorem 1 and the definition (1) that for each $j\in\mathcal{V}$:

$$\mathbb{E}\Big[\sum_{t=1}^{T}\sum_{i=1}^{N}\hat{f}_{i,t}(\mathbf{x}_{j,t})\Big] - \min_{\mathbf{x}\in(1-\xi)\mathcal{K}}\sum_{t=1}^{T}\sum_{i=1}^{N}\hat{f}_{i,t}(\mathbf{x}) \leq 3dNL_fR\left(1+\frac{4\rho(1+\sqrt{N})}{1-\rho}\right)\sqrt{T}. \tag{C.1}$$

By $\xi=\delta/r$ and a similar procedure as that of [4, Lemma 2], we can show that for any $\mathbf{x}\in\mathcal{K}$:

$$\sum_{t=1}^{T}\sum_{i=1}^{N}\frac{f_{i,t}(\mathbf{y}_{j,t}^1)+f_{i,t}(\mathbf{y}_{j,t}^2)}{2} - \sum_{t=1}^{T}\sum_{i=1}^{N}f_{i,t}(\mathbf{x})$$

$$\leq \sum_{t=1}^{T}\sum_{i=1}^{N}\hat{f}_{i,t}(\mathbf{x}_{j,t}) - \sum_{t=1}^{T}\sum_{i=1}^{N}\hat{f}_{i,t}((1-\xi)\mathbf{x}+NTL_f\delta(3+R/r). \tag{C.2}$$

This combined with (C.1) produces that

$$
\mathbb{E}\left[\sum_{t=1}^{T}\sum_{i=1}^{N}\frac{f_{i,t}(\mathbf{y}_{j,t}^1)+f_{i,t}(\mathbf{y}_{j,t}^2)}{2}\right] - \sum_{t=1}^{T}\sum_{i=1}^{N}f_{i,t}(\mathbf{x}^*)
$$
$$
\leq 3dNL_fR\left(1+\frac{4\rho(1+\sqrt{N})}{1-\rho}\right)\sqrt{T} + NTL_f\delta(3+R/r).
$$

Then we obtain the result by the selection of $\delta$. $\qquad\square$

*Proof of Theorem 6.* Recall that $\|\tilde{\mathbf{g}}_{i,t}\| \leq dL_f$ and $\|\nabla \hat{f}_{i,t}(\mathbf{x}_{i,t})\| \leq dL_f \triangleq G_f$. We can obtain from Theorem 2 and the definition (1) that for each $j \in \mathcal{V}$ :

$$
\mathbb{E}\Big[\sum_{t=1}^{T}\sum_{i=1}^{N}\hat{f}_{i,t}(\mathbf{x}_{j,t})\Big] - \min_{\mathbf{x}\in(1-\xi)\mathcal{K}}\sum_{t=1}^{T}\sum_{i=1}^{N}\hat{f}_{i,t}(\mathbf{x}) \leq \frac{Nd^2L_f^2}{2\alpha}\left(1+\frac{6\rho(1+\sqrt{N})}{1-\rho}\right)(1+\ln(T)).
$$

This combined with (C.2) produces that

$$
\mathbb{E}\left[\sum_{t=1}^{T}\sum_{i=1}^{N}\frac{f_{i,t}(\mathbf{y}_{j,t}^1)+f_{i,t}(\mathbf{y}_{j,t}^2)}{2}\right] - \sum_{t=1}^{T}\sum_{i=1}^{N}f_{i,t}(\mathbf{x}^*)
$$
$$
\leq \frac{Nd^2L_f^2}{2\alpha}\left(1+\frac{6\rho(1+\sqrt{N})}{1-\rho}\right)(1+\ln(T)) + NTL_f\delta(3+R/r)
$$

We then obtain the result by the selection of $\delta$. $\qquad\square$