[Reviews · NeurIPS 2020]

Review 1

Summary and Contributions: This paper studies distributed online optimization under a variety of feedback settings, and when the distributed agents are placed at the vertices of an Erdos-Renyi type graph. The problem is for all nodes to simultaneously choose a decision, and observe feedback corresponding to the loss function at that node (or some variant, such as a bandit loss function at the node). The goal for the agents however, is to choose actions so as to minimize the regret against the sum of all functions. The main challenge is that each agent can only receive feedback of its loss function (or the gradient depending on the feedback model), while the regret to be minimized is with respect to the entire network. In order to overcome the difficulty, agents exchange vectors with their neighbors that aid in collaboration. The precise model considered in the paper is that each agent exchanges a vector with each of its neighbor (on some base graph G), independently with probability at each time (after making a decision). The paper's key result demonstrates the regret any agent incurs, where the regret is defined as the total loss incurred by the agents decision and that of a fixed best decision in hindsight. The paper shows that for fixed (constant) network size and probability, the regret scaling in time matches that of a centralized setting. Furthermore, the paper characterizes the regret scaling with respect to network size and communication probability. The paper finds that the per agent regret is super-linear in N times the optimal regret in the centralized setting and inversely proportional to p. The key technical contribution is to develop and analyze algorithms that work when the information across the agents is heterogenous.

Strengths: The paper is reasonably well written, with the key ideas presented well. The technical content of the paper is presented nicely.

Weaknesses: One missing piece of discussion is to explain why the regret in N is super-linear. This implies, that as the number of agents N increases, the average (divided by N) regret scales as N^c for some 0 < c < 1. Thus, somehow, as the number of agents increases, the average regret also increases ? This can perhaps by explained by noting that as the number of agents increases, the heterogeneity of information across agents increases, thereby making regret minimization harder. A discussion on why this is super-linear in N can further improve explaining the key ideas and highlight the technical challenges posed by the problem setup.

Correctness: To the best of my knowledge, the proofs seem correct.

Clarity: Yes, it is quite well written. One suggestion will be to include a subsection highlighting the key differences in the proof compared to the centralized system. In other words, how is the information heterogeneity tackled in the proof, can be written as a proof sketch in the main paper.

Relation to Prior Work: Sufficient context of related work is given.

Reproducibility: Yes

Additional Feedback: Thanks for your rebuttal and response to my comments.


Review 2

Summary and Contributions: This is a theoretical paper that focuses on distributed online convex optimization. While previous work has focused on static and strongly connected, balanced networks, this work focuses on a classical random graph model -- Erdos-Renyi graphs. It shows that regret bounds of the same order as obtained in previous results for other settings hold for the Erdos-Renyi context, which additionally exposing how the probability of connections impacts the regret.

Strengths: This paper is the first to study distributed OCO in the context of a random graph model. It provides results that match known regret bounds for related contexts in three settings -- full gradient, one-point bandit, and two-point bandit feedback. Thus, it provides a complete characterization of the new model.

Weaknesses: Algorithms are each natural extension of prior work (as noted by the authors), so the novelty is in the analysis rather than the algorithmic formulation. While the proofs are interesting and non-trivial, it was difficult for me to understand which parts of the techniques were novel and which were adapted from related papers in the field. Clarity from the authors here would be helpful. The motivation provided for studying the Erdos-Renyi graph in this context was minimal. While it is, of course, a classical model and therefore of interest, it is difficult for me to imagine a place where it is the "right" model for a specific learning/control task. Clarity from the authors here would be helpful. The numerical examples are artificial. While the experiments use a real data set, there is no discussion about why the distributed graph model is relevant for the context and little exploration of impact of the properties of the graph.

Correctness: I have verified the included proofs.

Clarity: The paper is clear and well-organized. Additional context for motivation

Relation to Prior Work: The discussion of related work was clear and provided appropriate context for the paper.

Reproducibility: Yes

Additional Feedback: It would be interesting to include comments about what properties of the Erdos-Renyi model are crucial for the analysis. Is it natural or straightforward to consider extensions to related models like block stochastic models or the configuration model? or to models with heavy-tailed degrees such as preferential attachment? Thank you for your author response and it's answers to my questions. It was useful to hear elaboration on the motivation and applicability of the model. No Broader impact statement was included.


Review 3

Summary and Contributions: The paper studies the online convex optimization problem over an Erdos-Renyi network. In particular, the authors consider three regimes: with full gradient feedback, one-point bandit feedback, and two-point bandit feedback. The paper provides the algorithms in each case, as well as the regret bounds for both convex losses and strongly convex losses.

Strengths: To the best of my knowledge, the problem of online convex optimization over random graphs is novel. I think the model is interesting and potentially related to some real life distributed tasks. It seems the order of the regret bounds provided in the paper is fairly tight compared to the centralized version. The paper also provides communication complexity results. In general the results in this paper seem novel and relevant to the optimization community.

Weaknesses: I understand this is a theoretic work, but I think it would be great if the authors could provide some concrete examples before presenting the main results.

Correctness: The results seem correct.

Clarity: The paper is well written and easy to follow.

Relation to Prior Work: The model itself is novel. The paper includes some comparison between the new bounds and those in a centralized setting.

Reproducibility: Yes

Additional Feedback: What can the authors say about the relationship between the regrets and the graph topologies? For example, in Figure 4, it is shown that the performance on star graphs do not improve significantly as p grows (the other two cases). Are these trajectories related to some graph properties, say, minimum degree? ---- Post Rebuttal ---- It seems one of the main concerns of the reviewers is the motivation behind the model. I appreciate that the authors included some discussions in the rebuttal.


Review 4

Summary and Contributions: Post rebuttal: Many thanks to the authors for the detailed response and additional experiment results showing the impact of $N$ and $d$ in the regret bound. Unfortunately, I'm not knowledgeable enough to judge the tightness of the bound in terms of those factors. In case accepted, please provide those discussions in the final version of the paper. ---------------------- This paper provided the analysis of regret bounds for distributed OCO under convex or strongly convex losses, when the communication graph is an ER random graph. Regret bounds are derived to show dependences on network size, connection probability, and time horizon. Numerical results are provided to validate the theoretical claims.

Strengths: The theory part is the strength of the paper.

Weaknesses: 1. The paper is not very well-motivated. 2. The simulations are not convincing why the theoretical results would be important for practical applications.

Correctness: Claims seem correct to me, while empirical methodology is not sufficient enough to justify those claims.

Clarity: The logic of the paper is clear, but theoretical claims lack intuitive explanations. Typo: Algorithm 1, line 3, "Nodes" -> "Node". Typo: footnote in page 7 there is an extra space, but the url is invalid anyway.

Relation to Prior Work: From my impression reading this paper, the difference would be that the authors considered a distributed setting and a dynamic ER graph, while other works consider either centralized setting or fully connected and static communication graphs.

Reproducibility: Yes

Additional Feedback: Feedbacks for main text: 1. Motivation: the paper is not well-motivated. Neither the introduction nor the numerical results convinces me that the scenario studied in this paper would better fit into real-world applications than the scenarios considered in existing studies. For example, why the bodyfat dataset would be a good fit to the framework proposed by the authors and not other existing frameworks, is unclear. 2. The regret bounds show dependence on factors other than T, such as p and N, is the order of dependence optimal? 3. Numerical simulation dataset is not well-introduced. What are the features? Is it a reasonable assumption that (a_{i,t}, b_{i,t}) is revealed to node i only at time t? Also, the link to the dataset is broken. 4. In Figure 2, why do one- and two-point bandit feedback's performances worsen at the beginning? In Figures 3 and 4, the impact of p for one- and two-point bandit feedback is not clearly shown. It is also unclear how N or d impacts the performance. Furthermore, MReg is used in Figures 3 and 4 when the analysis is on expected regret. Therefore I don't think the claim that "the empirical finding matches theoretical results in Theorems 1-6" at the end of section 5 is true. 5. The authors might want to add discussions regarding scalability and practical computation issues. 6. The broader impact section is missing and the sty file is incorrect. Feedbacks for appendix: overall, the appendix is hard to follow as many steps of the proofs lack details needed (for me) to make connections. For example, the equation beneath (A.14), and (B.2). 1. In (A.3) and (A.6) left-hand side is missing a square. 2. In (A.8) \Omega is not defined. The intuitions of the quantities involved in Lemma 2 is not very clear and deserves more explanations. I think some quantities such as \Phi is also used later on, and it might be better to discuss the properties.

[Author Response · NeurIPS 2020]

We thank all the reviewers for the thoughtful feedback. We are encouraged that three reviewers voted to accept, and
acknowledge that the paper is well written and clearly presented ([R1],[R2],[R3],[R4]), "first to study distributed OCO
in random graphs" ([R2]), "novel" and "interesting and potentially related to some real life distributed tasks" ([R3]),
and has a "complete characterization" with a "nontrivial" and solid analysis ([R2],[R3],[R4]). We respond to main
comments below and will address all feedback in the main paper.

**Motivation and example ([R2], [R3], [R4]).** We consider distributed OCO over Erdos-Renyi graphs motivated by
the following. (1) Distributed optimization and learning over random graphs has been widely studied since it is a
validated model in practical communication networks, e.g., for modeling package losses in wireless communications.
(2) Erdos-Renyi graphs give us information about complex systems which exist in the real world. The Internet or
social networks provide the example at the moment, and it is equally plausible to think about traffic flows, electrical
systems or interacting biological processes ([1]). (3) It has been found that Erdos-Renyi random graphs can outperform
fully connected graphs in some distributed training tasks ([2]). Moreover, in the dense network, we might want to
avoid communicating along each edge per iteration to decrease communication, for which Erdos-Renyi graphs allow a
more refined tradeoff between computation and communication costs. (4) Erdos-Renyi graphs allow us to intuitively
have a deep theoretical characterization of how the graph topology and connectivity probability influence the OCO
network regrets, and paves the way to study distributed OCO over random graphs with a power-law degree distribution
by preferential attachment. A prominent motivating example is the distributed online learning through random social
interactions for exploiting the streaming but private healthy data generated from wearable personal tracking device
([3]), which also motivates our simulation study with *bodyfat* dataset.

**Discussions of theoretical results on factors, such as $N$ ([R1], [R4]), graph topology ([R3]), and $p$ ([R4]).** (1) The
average regret increases with $N$, possibly because the increasing node number would increase the nodes' information
heterogeneity and make the network regret minimization harder as Reviewer 1 suggested. Compared with the previous
work by Lobel and Ozdaglar for distributed optimization over random networks, which provided bounds that grew
exponentially in $N$, we achieved polynomial scaling on network size. (2) The derived regrets showed the inverse
dependence on the spectral gap of the expected network, which is quite natural since it is well-known to determine the
mixing rates in random walks on graphs, and the information propagation over Erdos-Renyi graphs is closely tied to the
random walk on the expected network. (3) It remains open what is the optimal order of dependence on factors $N$, $d$,
and $p$ in the regrets. (4) We give a simulation study to show how $N$ and $d$ impact the algorithm performance.

**Technical challenges ([R1],[R2])** By making use of the 9th page allowed in the camera-ready version, we will elaborate
on as many proof outlines as possible in the main body. The proof novelty induced from the Erdos-Renyi graphs lies in
showing that the expected consensus error depends on the inverse spectral gap of the expected network.

**Simulation Justification([R2],[R4])** We adopt *bodyfat* dataset in numerical studies since distributed OCO over Erdor-
Renyi graphs is a practical and preferred learning framework for exploiting personal healthy data from wearable
personal digital device or personal tracking device. Concerns on scalability and privacy make the distributed learning a
preferable method than centralized ones, while the streaming of the data entails online learning.

**Other discussions in simulations [R4]** At the beginning steps of Figure 2, there is not enough accumulated data to
adapt a good solution in the one-point bandit case, hence the performance can get deteriorated in the first few steps.
This is accordant with theoretical results that the time averaged regret goes to zero as $T$ goes to infinity. In Figure 3
and Figure 4, the regret is decreasing with the increment of $p$, while the decreasing magnitude depends on the graph
topology. MReg is the maximal of expected regret of all nodes, hence is taken as a representative performance index in
simulations.

[1]: Franceschetti & Meester. Random Networks for Communication: From Statistical Physics to Information Systems. Cambridge
University Press. (2008) doi:10.1017/CBO9780511619632.

[2]: Adjodah, Dhaval, et al. "Communication topologies between learning agents in deep reinforcement learning." Arxiv preprint at
arXiv:1902.06740.

[3]: Brisimi, Theodora, et al. "Federated learning of predictive models from federated electronic health records." International
Journal of Medical Informatics 112 (2018): 59-67.


[Meta-Review · NeurIPS 2020]

The paper considers the problem of distributed online convex optimization over Erdos Renyi radnom graphs and provides analysis of online gradient descent algorithm in the full information setting and one point and two point bandit query models. Both convex Lipschitz and strongly convex losses are considered. For full information case the regret bound is shown to match the rates for classical online gradient descent in terms of Horizon T. For two point bandit query the rates are shown to match the optimal rates in classical case as well. For one point query a possibly sub-optimal rate of T^3/4 and T^2/3 are shown for Lipschitz and strongly convex setting. Overall the reviewers found the work interesting. One main concern that was raised was about motivation of the setting which the authors seemed to have answered somewhat satisfactorily. For the one point query setting, in the classic setting (for now say convex Lipschitz set up), one can get sqrt{T} regret bound using Bernoulis kernel based on Bubeck et al. Is there any hope of distributing this, I dont see much discussion about this in the paper.